# Clinical Manifestations of Human Exposure to Fungi

**DOI:** 10.3390/jof9030381

**Published:** 2023-03-21

**Authors:** Manuela Oliveira, Diana Oliveira, Carmen Lisboa, José Laerte Boechat, Luís Delgado

**Affiliations:** 1i3S—Instituto de Investigação e Inovação em Saúde, Universidade do Porto, Rua Alfredo Allen 208, 4200-135 Porto, Portugal; 2Ipatimup—Instituto de Patologia e Imunologia Molecular da Universidade do Porto, Rua Júlio Amaral de Carvalho 45, 4200-135 Porto, Portugal; 3CRN—Unidade de Reabilitação AVC, Centro de Reabilitação do Norte, Centro Hospitalar de Vila Nova de Gaia/Espinho, Avenida dos Sanatórios 127, 4405-565 Vila Nova de Gaia, Portugal; 4Serviço de Microbiologia, Departamento de Patologia, Faculdade de Medicina do Porto, Alameda Prof. Hernâni Monteiro, 4200-319 Porto, Portugal; 5Serviço de Dermatologia, Centro Hospitalar Universitário de São João, Alameda Prof. Hernâni Monteiro, 4200-319 Porto, Portugal; 6CINTESIS@RISE—Centro de Investigação em Tecnologias e Serviços de Saúde, Faculdade de Medicina, Universidade do Porto, Alameda Prof. Hernâni Monteiro, 4200-319 Porto, Portugal; 7Serviço de Imunologia Básica e Clínica, Departamento de Patologia, Faculdade de Medicina, Universidade do Porto, Alameda Prof. Hernâni Monteiro, 4200-319 Porto, Portugal; 8Laboratório de Imunologia, Serviço de Patologia Clínica, Centro Hospitalar e Universitário de São João, Alameda Prof. Hernâni Monteiro, 4200-319 Porto, Portugal

**Keywords:** allergens, allergy, fungal diseases, fungi, hypha, immunodeficiency, spores

## Abstract

Biological particles, along with inorganic gaseous and particulate pollutants, constitute an ever-present component of the atmosphere and surfaces. Among these particles are fungal species colonizing almost all ecosystems, including the human body. Although inoffensive to most people, fungi can be responsible for several health problems, such as allergic fungal diseases and fungal infections. Worldwide fungal disease incidence is increasing, with new emerging fungal diseases appearing yearly. Reasons for this increase are the expansion of life expectancy, the number of immunocompromised patients (immunosuppressive treatments for transplantation, autoimmune diseases, and immunodeficiency diseases), the number of uncontrolled underlying conditions (e.g., diabetes mellitus), and the misusage of medication (e.g., corticosteroids and broad-spectrum antibiotics). Managing fungal diseases is challenging; only four classes of antifungal drugs are available, resistance to these drugs is increasing, and no vaccines have been approved. The present work reviews the implications of fungal particles in human health from allergic diseases (i.e., allergic bronchopulmonary aspergillosis, severe asthma with fungal sensitization, thunderstorm asthma, allergic fungal rhinosinusitis, and occupational lung diseases) to infections (i.e., superficial, subcutaneous, and systemic infections). Topics such as the etiological agent, risk factors, clinical manifestations, diagnosis, and treatment will be revised to improve the knowledge of this growing health concern.

## 1. Introduction

Biological particles, along with inorganic gaseous and particulate pollutants, constitute an ever-present component of the atmosphere and surfaces. Among these bioparticles are fungal spores and hyphae [1,2] that colonize almost all terrestrial [3] and aquatic ecosystems [4,5], and even the inside and outside of the human body [6,7,8]. This diverse kingdom is composed of approximately 150,000 known species, and it is estimated that nearly a thousand times more species are yet to be described [9].

Fungal spores rarely exceed 20 µm in diameter, ranging from 3 to 8 µm [10]. Smaller fungal spores can travel greater distances than larger spores [11]. Due to their small size, most allergenic fungal spores can penetrate lower airways of sensitized individuals [12], acting as sub-pollen particles from rupturing pollen grains [13].

The following sections will address the several clinical manifestations associated with exposure to fungal spores. The first section describes diseases related to the host’s adverse immune responses, while the second includes illnesses related to direct infection of the host by fungal pathogens.

## 2. Allergic Fungal Diseases

Exposure to fungal particles (i.e., spores, hyphae, and metabolites) is associated with several allergic diseases, such as allergic bronchopulmonary aspergillosis (ABPA; estimated globally in approximately 4,800,000 adults), severe asthma with fungal sensitization (SAFS; some 6,000,000 persons), and allergic fungal rhinosinusitis (AFRS; approximately 12,000,000 individuals) [14].

Allergic fungal diseases can be divided into two groups. The first group includes allergic responses to aeroallergens (e.g., *Alternaria* spp. and *Cladosporium* spp.) that present airborne concentrations overlapping seasonal disease symptoms of acute allergic exacerbations. The second group includes an allergic response to thermotolerant filamentous fungal genera (e.g., *Aspergillus* spp. and *Penicillium* spp.) that not only act as aeroallergens but can also germinate in the airway, colonizing the lung and leading to a persistent allergenic stimulus that damages lung tissue [15].

Worldwide fungal allergy prevalence has been estimated to be between 3 and 10% [1], being higher for allergic sensitization when evaluated by skin prick tests in allergic (19–45%) and asthmatic patients (80%) [16,17,18,19].

Fungal spores play a significant role as aeroallergens in asthma development, exacerbation, and severity. About 120 individual fungal allergens from thirty-one mold genera have been identified (http://www.allergen.org/, accessed on 16 January 2023) (Appendix A). Among the most well-known fungal sensitizing agents for asthmatic patients are *Alternaria alternata*, *Aspergillus fumigatus*, and *Cladosporium herbarum* [20].

The following sub-section provides further information concerning allergic fungal diseases, such as ABPA, SAFS, thunderstorm asthma (TA), AFRS, and occupational lung diseases (Table 1 and Table 2).

### 2.1. Allergic Bronchopulmonary Aspergillosis (ABPA)

In susceptible hosts (mainly patients with asthma or cystic fibrosis), repeated inhalation of *A. fumigatus* spores, followed by respiratory tract colonization, triggers an immune hypersensitivity reaction [21]. This reaction is mediated by an immunoglobulin E (IgE), with possible involvement of immunoglobulin G (IgG)-mediated immune complex and cell-mediated hypersensitivity reactions [22,23].

ABPA affects between 1 and 15% of cystic fibrosis patients and 2.5% of asthma patients, corresponding to nearly 4.8 million people worldwide [22]. This progressive disease has no gender predilection and is characteristic of young and middle-aged adults with persistent allergic asthma, being frequently misdiagnosed and poorly managed with limited favorable prognosis [24,25].

ABPA early symptoms are commonly confounded with symptoms of the underlying disease. With the progression of the disease, there is chronic bronchial inflammation accompanied by eosinophilia, airway remodeling, and bronchiectasis (leading to the development of areas of parenchymal scarring). Bronchi may present mucus plugs containing fungal hyphae, fibrin, Charcot–Leyden crystals, and Curschmann spirals [26]. Systemic symptoms such as fever, malaise, and weight loss are common in ABPA and may raise suspicion in patients with asthma (in whom such symptoms are not commonly seen) or cystic fibrosis.

ABPA diagnosis is based on an interaction between clinical, biological, and radiological findings following diagnostic criteria. Since Hinson’s discovery in 1952, ABPA diagnostic criteria have been frequently revised, as delaying ABPA diagnosis predisposes patients to more severe and prolonged complications [27,28,29]. There are not internationally widely accepted criteria for the diagnosis of ABPA. Rosenberg and Patterson proposed the first diagnostic classification in 1977 [30], later modified by Schwartz and Greenberger in 1991 [31]. The main contribution of the latter was the division of ABPA according to the presence of central bronchiectasis (ABPA-CB) or its absence (ABPA-S, serologic).

The most recent guidelines proposed by the International Society for Human and Animal Mycology (ISHAM) in 2013 include mandatory (two points that must be present) and other criteria (three points, two of three must be present). The mandatory criteria are immediate cutaneous reactivity to *A. fumigatus* or elevated IgE levels directed against this fungus and high total IgE levels of more than 1000 IU/mL. The other criteria are detecting IgG antibodies against *A. fumigatus*, pulmonary opacities observed on chest radiograph, and eosinophil counts above 500 cells/μL in steroid-naïve patients [32,33].

ABPA treatment aims to control asthma or cystic fibrosis symptoms, prevent or treat pulmonary ABPA exacerbations, reduce pulmonary inflammation, and reduce the progression to central bronchiectasis or fibrotic disease [25,34]. Oral corticosteroids (e.g., prednisone) are the primary therapeutical approach, and serum total IgE monitors disease activity [24]. Treatment regimens with prednisone are variable, but the most used is a low dosage (0.5 mg/kg/day) every day for two weeks, then on alternate days for three months.

In corticosteroid-dependent ABPA patients, antifungal agents (mainly azoles, e.g., itraconazole or voriconazole) effectively control symptoms and reduce the need for systemic corticosteroids. Antifungals also had a positive impact on biomarkers, and radiological pulmonary infiltrates. However, the evidence supporting these results (mainly in terms of treatment duration and shortage of controlled studies) and the presence of adverse effects means the adjunctive azole therapy in ABPA should be evaluated on a case-by-case basis [25,34,35,36].

Several biological drugs targeting the inflammatory allergic component of the disease (e.g., anti-IgE (omalizumab), anti-IL5 (mepolizumab and benralizumab), and anti-IL4R (dupilumab)) have been recently tested in clinical practice [37,38,39,40]. Most studies evaluated the use of omalizumab, and a recent meta-analysis demonstrated the clinical benefits of this biologic [41]. The benefits of using biologics appear to be greater in patients with ABPA and asthma than in patients with cystic fibrosis [34]. However, robust data are lacking, and more controlled trials are needed before anti-IgE or other biologic therapies can be recommended to manage ABPA [42].

Finally, other fungi, such as *Aspergillus flavus*, *Aspergillus niger*, *Penicillium* spp., and *Schizophyllum commune*, can cause pathologies similar to ABPA, designated as allergic bronchopulmonary mycosis (ABPM) [43,44].

### 2.2. Severe Asthma with Fungal Sensitization (SAFS)

Bronchial asthma, an airway inflammatory disease characterized by variable airflow limitation and airway hyperresponsiveness, is worsened by numerous extrinsic factors. Among these factors are aeroallergens, including fungal spores and hyphae. Severe asthma impacts 5 to 10% of total asthma patients, presenting limitations on their daily activities and even being fatal [45].

Allergic asthma is associated with type I hypersensitivity (positive serum allergen-specific IgE and/or skin-prick test (SPT) to common aeroallergens). Although asthma is determined by genetic and epigenetic factors, sensitivity to indoor and outdoor aeroallergens acts as a trigger in individuals genetically predisposed to developing allergic diseases (atopy) [46]. Clinical symptoms include chest tightness, dyspnea, cough, and wheezing [47].

Severe asthma with fungal sensitization (SAFS) can be addressed as an intermediate disease between allergic asthma and allergic bronchopulmonary aspergillosis [47]. Fungal spore exposure and IgE sensitization have been associated with worsening asthma symptoms, reduced lung function, increased hospital admissions, and asthma-related mortality in children and adults [47]. Nearly 20% of patients with severe asthma have positive SPT or in vitro tests for fungus-specific IgE, and fungal sensitization is associated with increased asthma severity [45]. In some SAFS patients, several radiological abnormalities observed on high-resolution chest computed tomography (CT) (i.e., minor degrees of bronchiectasis, upper lobe fibrosis, tree in bud, and consolidation) have been reported [48]. The criteria proposed for SAFS diagnosis include the occurrence of severe asthma (which is symptomatic and uncontrolled asthma despite the treatment with high doses of inhaled corticosteroids plus a second controller and/or systemic corticosteroids) [49], evidence of IgE sensitization to molds (SPT or fungal-specific IgE), mainly to thermotolerant filamentous fungi (*Aspergillus* spp. or *Penicillium* spp.), and exclusion of ABPA (Figure 1) [16,47].

The initial treatment of patients with SAFS should be similar to that of patients with severe asthma: asthma education; treatment optimization (check and correct inhaler technique and adherence, switch to inhaled corticosteroid-formoterol maintenance and reliever therapy); non-pharmacological interventions (e.g., smoking cessation, weight loss, exercise); comorbidities treatment and non-biologic add-on therapy (e.g., long-acting muscarinic antagonists, leukotriene modifier/leukotriene receptor antagonist, if not used) [50]. Nevertheless, SAFS appears to be associated with more treatment-refractory asthma. When the treatment response is partial or absent, antifungal drugs and biologics have been considered in treating this condition. The improvements with antifungal agents such as itraconazole are modest [51]. A recent study showed that omalizumab (anti-IgE) is an effective add-on therapy for severe asthma with fungal sensitization, with treatment over 24 months [52]. A real-world study highlights the potential clinical utility of mepolizumab or benralizumab (anti-IL-5/5R therapy) in patients with SAFS, with improvements in exacerbation frequency and patient-reported outcomes, and reduction in maintenance dose of oral corticosteroids over a 48-week treatment period [53]. Despite the above, the role of itraconazole and type-2-targeted biologicals in managing SAFS requires further evaluation before being considered as a first-line treatment for this specific asthma phenotype [54].

### 2.3. Thunderstorm Asthma (TA)

Thunderstorm asthma can be defined as the increase in acute asthma exacerbations after thunderstorms affecting individuals close to each other and the associated storm front [55]. Some patients report the onset of symptoms occurring within minutes to hours of the preceding wind gusts. These rare events depend on the conjugation of specific populations and meteorological and environmental conditions [55]. The symptoms overlap with common asthma symptoms such as coughing, shortness of breath, wheezing, and respiratory distress. Some patients are admitted to hospital facilities for proper medical care, but fatalities are rarely observed, and symptoms frequently improve with the optimization of inhaled therapies [55].

Although poorly predicted, sporadic, and infrequent, these events mainly occur in late spring or early summer periods, coinciding with the seasons of higher circulating aeroallergen counts. Among these aeroallergens are pollen grains (mainly grass but also weed and tree pollens) and fungal spores (mainly *Alternaria* spp., *Cladosporium* spp., Diatrypaceae, *Didymella exitialis*, *Phaeospaeria nigrans*, and *Sporobolomyces* spp.) [56]. It has been proposed that electrical charges are responsible for pollen and spores’ fragmentation, transforming larger particles to breathable sizes (e.g., from >35 µm to <3 µm) that reach the lower respiratory tract, and ionization of exposed allergens that promote longer adherence to the mucous membranes of lower airways. These two conditions may elicit allergic reactions in predisposed individuals [57,58].

Several risk factors can be associated with individual susceptibility to TA [55]. Among them is a history of allergen sensitization, being outdoors or indoors with open windows [24], seasonal allergic rhinitis [59], age between 20 and 50 years [60], being male [61], and belonging to Asian or Indian ethnic groups [62]. In more severe cases of thunderstorm asthma, requiring intensive care admission or resulting in death, individuals already presented a diagnosis of asthma [59].

### 2.4. Allergic Fungal Rhinosinusitis (AFRS)

Initially considered a rare disease, the burden of fungal rhinosinusitis is increasing because of the rising trend in life expectancy, medical advancements with invasive interventions, the misusage of prescription drugs (e.g., corticosteroids, antibiotics), and the number of patients with primary or secondary immunodeficiencies (e.g., underlying conditions such as diabetes mellitus, malignancies) [63].

Allergic fungal rhinosinusitis is a noninvasive form of sinus inflammation and one of the most common forms of fungal sinus disease, corresponding to 6–9% of all chronic rhinosinusitis cases. AFRS typically occurs in younger immunocompetent atopic patients (21–33 years) with sinus fungal colonization and is more common in geographic regions characterized by warm temperatures and high humidity [64]. Specific risk factors associated with AFRS are exposure to fungi, history of classic allergic rhinitis whose symptoms become progressively severe and unresponsive to antihistamines and inhaled nasal corticosteroid, in situ fungal growth, and underlying respiratory disease (rhinosinusitis or asthma) [65]. The prevalence of asthma in AFRS patients is lower than that reported in other chronic rhinosinusitis with nasal polyps (CRSwNP) subtypes, and gender predominance varies between reports [64].

In these patients, even in the absence of fungal colonization, type I (IgE-mediated reaction to an antigen) and type III (IgG-mediated antigen-antibody complex formation) hypersensitivity reactions to fungal antigens are observed, inducing the production of ‘eosinophilic’ mucin with the enrolment of several inflammatory mediators (e.g., Th2 cytokines, such as interleukin (IL)-4, and IL-5) [63]. ‘Eosinophilic’ or ‘allergic’ mucin is a trademark of AFRS, macroscopically defined as a thick, tenacious mucus, highly viscous in consistency and light tan to brown (peanut-butter-like appearance) or dark green. Microscopically, it is characterized by lamellated aggregates of dense inflammatory cells, mainly eosinophils and Charcot–Leyden crystals, and fungal hyphae (Dematiaceous fungi and *Aspergillus* spp. are commonly identified) [66,67]. *Staphylococcus aureus* routinely colonizes the sinuses of patients with AFRS, and the presence of a symbiotic relationship between the fungi and *S. aureus* has been proposed [67,68]. Regarding biomarkers, patients with AFRS have very high total and specific IgE levels for fungi and relatively normal serum eosinophil levels compared with other patients with CRSwNP [64].

Symptoms of AFRS include hyposmia/anosmia, inflammation in the nose and sinuses, nasal congestion, rhinorrhea, posterior pharyngeal drainage, pain, pressure in the sinus area (pain when touching the cheeks or forehead), and sinus headache. The mucin expansion within the sinuses leads to bone and tissue expansion, resulting in some cases to changes in facial appearance, such as facial asymmetry and proptosis [67].

The current guidelines for AFRS diagnosis are based on major (obligatory) and minor (complementary) criteria [67,69,70,71]. The primary criteria include: (1) type I hypersensitivity to fungi (confirmed by skin test or IgE ImmunoCAP); (2) nasal polyposis; (3) characteristic CT scan findings (i.e., unilateral or bilateral opacification and expansion of multiple paranasal sinuses with centrally hyperdense content); (4) presence of eosinophilic mucin without invasion into the sinus tissue, and (5) presence of fungal spores or hyphae (e.g., *Alternaria* spp., *A. fumigatus*, *A. flavus*, *A. niger*, *Bipolaris spicifera*, *Curvularia lunata*, and *Cladosporium* spp.) on sinus contents removed during the surgery. Minor criteria include the unilateral predominance of disease, bone erosion revealed, fungal growth in cultures, Charcot–Leyden crystals in surgical specimens, and serum eosinophilia [67,69,71]. Because patients with eosinophilic CRS can have eosinophilic mucin and atopy, AFRS diagnosis requires the presence of all five major criteria [66].

The therapeutical approach includes medical and surgical procedures. Functional endoscopic sinus surgery (FEES) is essential to eliminate as many fungi and allergic mucin as possible and to open all involved sinuses, facilitating the delivery of topical medication [72]. Adjuvant medical therapy is essential for the successful treatment of AFRS. Topical and oral corticosteroids are the basis of medical therapy for AFRS, used for perioperative optimization and long-term disease control postoperatively [73]. In some patients, long-term treatment with systemic corticosteroids is necessary to maintain clinical remission, with risk-benefit assessment being necessary on a case-by-case basis [64].

Although systemic and topical antifungals have been frequently used in patients with AFRS, there is currently insufficient evidence to recommend for or against antifungal therapy in AFRS [67,74].

Despite the central role of type I hypersensitivity to fungi in the pathogenesis of AFRS, allergen immunotherapy is another controversial area of AFRS treatment. Given the limitations in the currently available evidence, a systematic review regarding immunotherapy in AFRS patients was unable to recommend either for or against this treatment modality [75]. However, in recalcitrant AFRS, some studies report better quality of life, diminished corticosteroid requirements, and revision surgeries in patients using fungal immunotherapy compared with controls [76,77]. In conclusion, fungal immunotherapy remains an option for treatment in a selected case rather than a recommendation.

As type 2 inflammation is a hallmark of AFRS, biologics targeting type 2 inflammatory mediators (e.g., IgE, IL-4, IL-5, and IL-13) are a potential treatment option for AFRS patients. It should be noted, however, that patients with AFRS were not included in pivotal Phase III studies of biologics approved for treating CRSwNP (dupilumab, omalizumab and mepolizumab) [78,79,80]. Despite this, case reports and smaller prospective studies suggest the effectiveness of dupilumab, omalizumab, and mepolizumab in AFRS patients, positioning these biologics as an add-on therapy for refractory or steroid-dependent cases [64].

### 2.5. Occupational Lung Diseases

Several types of hypersensitivity pneumonitis (HP), or extrinsic allergic alveolitis, have been associated with occupational exposures, hobbies, recreational activities, or contaminated air systems (Table 2). Hypersensitivity pneumonitis is an immunologic lung disorder affecting pre-sensitized subjects. The chronic or repeated inhalation of allergens induces a hypersensitivity reaction with granulomatous inflammation in the distal bronchioles and alveoli [81]. Inhalation of the inciting antigen(s) may occur in the workplace, and up to 5–15% of the workers may develop HP [82].

Although HP symptoms may develop a few days to weeks after allergen exposure, most HP cases occur after months or years of continuous or intermittent inhalation [82]. The clinical presentation and progression can be highly variable, depending on the nature of the inciting antigen, exposure intensity, and duration, and probably also of the individual immunogenetic background [83,84]. Hypersensitivity pneumonitis may have an acute presentation with non-specific symptoms, such as fever, chills, myalgia, headache, coughing, chest tightness, dyspnea, and leukocytosis, or may also be insidious, developing and worsening over months to years into a chronic form characterized by progressive exertion dyspnea and dry cough. In some patients with recurrent acute episodes, a chronic obstructive pulmonary disease with centrilobular emphysema has been described [82,83]. According to recent clinical practice guidelines, a consensus HP classification in fibrotic and non-fibrotic/inflammatory types (lacking radiographic or histologic fibrosis) is considered more consistent with its distinctive clinical course and outcomes [85,86]. The non-fibrotic or inflammatory subtype may have a benign course, while fibrotic HP often presents a progressive fibrotic phenotype similar to other fibrotic interstitial lung diseases [84].

Hypersensitivity pneumonitis diagnostic criteria, recently proposed in guidelines endorsed by respiratory societies, are based on the integration of exposure identification, high-resolution computed tomography (HRCT) findings, and bronchoalveolar (BAL) lymphocytosis. In selected cases, histopathology of lung biopsies can increase diagnostic accuracy [85,86,87]. A systematic investigation is needed to identify the inciting antigen(s) in the patient’s work and/or domestic environment. An industrial hygienist may be needed to identify and obtain appropriate samples from potential sources of exposure. A patient suspected of having HP should be referred to specialist care. As proposed in other interstitial lung diseases, the diagnostic workup will benefit from a consensus-based multidisciplinary discussion panel [88].

Prevention of exposure to the inciting antigen is the cornerstone of HP management because maintaining antigen inhalation has prognostic implications, with a possible disease progression and decreased survival rate. High-intensity and more extended periods of antigen exposure, advanced age, smoking, and a radiological and/or histological pattern of fibrosis are associated with a poor prognosis [83,84]. In non-fibrotic HP, the patient is anticipated to improve with antigen removal and, if clinically needed, systemic corticosteroid therapy. Patients with fibrotic HP usually do not normalize with an end to exposure only, and systemic immunomodulatory and/or anti-fibrotic therapy is commonly required. Nevertheless, some may develop progressive pulmonary fibrosis despite antigen eviction, being lung transplantation the only valuable treatment option [82,89].

Farmer’s lung disease (FLD) is caused by inhaling microorganisms growing on hay or grain stored in high-humidity conditions in the agricultural workplace. Worldwide, FLD affects between 0.5 and 3% of farmers and is associated with higher mortality rates. The etiological agents are thermophilic actinomycetes (e.g., *Saccharopolyspora rectivirgula*, *Thermoactinomyces vulgaris*, *Thermoactinomyces viridis*, and *Thermoactinomyces sacchari*) and fungi (e.g., *Alternaria* spp., *A. fumigatus*, *Botrytis* spp., *Penicillium brevicompactum*, and *Penicillium olivicolo*). Farm-related activities such as handling damp hay, opening bales for feeding livestock, and moldy threshing grains mainly contribute to exposure [89,90].

Mushroom worker’s lung disease has been reported in workers who inhaled spores from different species such as button (*Agaricus bisporus*) [91], Buna-shimeji (*Hypsizygus tessellatus*) [92], shiitake (*Lentinula edodes*) [93], and oyster mushrooms (*Pleurotus ostreatus*) [94]. Spores are exposed in spawning sheds while cultivating and handling mushrooms [95].

Suberosis (also known as cork handler’s disease or cork worker’s lung) occurs in cork workers, and is mainly reported in Portugal [12,96] and Spain [97]. In Portugal, 9–19% of cork workers may develop suberosis [98]. During cork manufacturing, workers are exposed to cork contaminated with *Penicillium glabrum* (formerly *Penicillium frequentans*), the primary etiological agent of suberosis. During storage, cork is colonized by other fungi, such as *Chrysonilia sitophila*, *A. fumigatus* or *Mucor* spp. [12,97].

Maple bark disease (or maple bark stripper’s disease) occurs when the workers inhale fungal spores from *Cryptostroma corticale*. This fungus causes sooty bark disease under the bark of maples and sycamore trees, which becomes airborne when stripping bark from logs [99,100].

Sequoiosis is caused by inhaling redwood (*Sequoia sempervirens*) sawdust from moldy redwood bark containing spores of various fungi, such as *Graphium* spp., *Pullularia* spp., *Aureobasidium pullulans*, among other fungi [101,102].

Wood pulp workers’ disease results from prolonged exposure to logs contaminated with *Alternaria* during the manufacture of wood pulp [103].

The exposure to *Botrytis cinerea* spores provokes wine growers’ or berry sorters’ lungs. This disease affects workers collecting grapes during vintages or handling them in the cellars during wine production [104].

Malt worker’s lung is caused by inhaling spores from *Aspergillus clavatus*, *Penicillium granulatum*, *Penicillium citrinum*, and *Rhizopus stolonifer* when handling contaminated germinating barley grains on the malt floors [105,106,107,108].

Baker’s lung disease is induced by inhaling *A. fumigatus* spores in workers from bakeries who handle flour (e.g., maize and oat) [109].

Cheese workers’ lung is caused by several fungi such as *Penicillium roqueforti*, *Penicillium casei*, *Penicillium viridicutum*, *A. fumigatus*, *A. niger*, and *Aureobasidium pullulans*. This disease affects workers cleaning moldy cheeses in cheese manufacturing facilities [110,111,112].

Salami brusher’s disease occurs because of the inhalation of spores from *P. glabrum*, *Penicillium camemberti*, *Penicillium nalgiovense*, *A. fumigatus*, and *Cladosporium* spp. In food processing facilities, workers responsible for removing the excess of white mold growing on the salami’s surface use a manual wire brush that aerolizes these bioparticles [113,114].

Tobacco workers’ lung is induced by exposure to *A. fumigatus* spores. It affects workers exposed to tobacco leaves and molds in the damp environment of tobacco cultivation fields and manufacturing facilities [115].

Peat moss worker’s lung, provoked by *Penicillium citreonigrum* and *Monocillium* spp., occurs in packaging plants when workers inhale the dust of contaminated peat moss [116].

The inhalation of *Mucor stolonifera* spores induces paprika slicer’s lung in workers handling moldy paprika pods during paprika slicing [83].

The above-described cases report occupational lung diseases due to exposure in workplaces. However, exposure can also occur at home, frequently escaping early recognition. Several HP diseases are associated with exposure to air through humidifiers, heating, and ventilation systems used in domestic and work settings. These types of equipment are continuously run without periodical cleaning or proper maintenance and accumulate spores from *Aspergillus* spp., *Cladosporium* spp., *Penicillium* spp., *A. pullulans*, *Cephalosporium* spp., *Mucor* spp. [117,118].

Summer-type HP occurs in response to the inhalation of arthroconidia from *Trichosporon cutaneum*, *Trichosporon asahii*, and *Trichosporon mucoides* that contaminate home environments (e.g., tatami mats, pillars, and beds) during the summer season in Japan [119,120].

Moreover, HP can result from exposure associated with recreational activities or hobbies. Sax lung disease has been reported in musicians, predominantly saxophone, trombone, and bagpipe players, who become contaminated through the mouthpiece of the wind instruments. In such cases, instruments are contaminated with *Ulocladium botrytis*, *Phoma* spp., *Fusarium* spp., *Penicillium* spp., *Cladosporium* spp., *Cryptococcus* spp., *Paecilomyces variotti*, *Rhodotorula mucilaginosa*, and *Trichosporon mucoides* [121,122,123,124].

## 3. Association of Immunodeficiencies and Fungal Diseases

Invasive fungal infections are associated with high morbidity and mortality. These infections primarily occur in patients with secondary or acquired immunodeficiency (e.g., patients subjected to hematopoietic stem cell or solid organ transplantation, chemotherapy and immune-modulatory treatment for malignancies, hospitalization in an intensive care unit (ICU), or presenting AIDS or autoimmune disorders). More rarely, patients with primary immunodeficiency diseases (PIDs, also known as inborn errors of immunity), mainly seen in children and young adults, are genetically predisposed to recurrent, unusual, prolonged, and more severe invasive fungal infections [125,126,127,128].

*Candida* spp., *Aspergillus* spp., *Cryptococcus* spp., and *Pneumocystis* spp. are responsible for more than 90% of reported deaths due to invasive fungal disease (IFD) [129]. These infections are often associated with iatrogenic immunosuppression [130].

Invasive candidiasis is a disease of critically ill hospitalized patients and originates from the patient´s flora following iatrogenic breaks of the skin or mucosal barrier. Using an intravenous catheter is the leading risk factor for candidaemia, the most common manifestation of invasive candidiasis [129].

Invasive pulmonary aspergillosis, secondary to inhalation of *Aspergillus* spores, is the main problem in immunocompromised hosts, accounting for >85% of invasive fungal diseases in humans. Risk factors for invasive aspergillosis include hematological malignancy, prolonged severe neutropenia, ICU patients (mainly chronic obstructive pulmonary disease and severe alcoholic liver disorders) and some forms of primary immunodeficiency [129].

Cryptococcosis is the most prevalent fatal fungal disease worldwide [131]. Cryptococcal meningitis is the most common manifestation, presenting as an opportunistic infection mainly in HIV-positive patients with low CD4 cell counts or iatrogenically immunosuppressed ones; immunocompetent patients are more likely to have the pulmonary disease [129].

In the pre-antiretroviral therapy era, *Pneumocystis jirovecii* pneumonia (PCP) was the most common opportunistic infection in HIV-positive patients. Nowadays, iatrogenically immunocompromised patients (malignancy, transplantation or rheumatological disease) are the leading risk group for PCP [129]. Usually, it is unclear if the disease results from recent reinfection or the reactivation of latent infection [132].

Primary immunodeficiency diseases include a diverse group of diseases caused by inherited expression defects in more than 485 genes, generally resulting in a reduced or absent function in one or more components of the immune system [133,134,135,136]. The number of identified PIDs is increasing with the developments of genetic detection methods, such as next-generation sequencing. The 476 described PIDs are grouped according to the immune system malfunction component and commonly divided into disorders of adaptive immunity (e.g., T-cell immunodeficiency, B-cell immunodeficiency, combined immunodeficiencies and severe combined immunodeficiency), innate immunity (e.g., phagocyte defects and complement defects), and immune dysregulation (e.g., autoimmune polyendocrinopathy–candidiasis–ectodermal dystrophy, Chediak Higashi Syndrome) [134,137,138].

Severe combined immunodeficiency (SCID), some combined immunodeficiencies (CIDs) (e.g., autosomal recessive (AR) Hyperimmunoglobulin E syndrome (HIES), Nuclear factor-kB essential modulator (NEMO) deficiency) and PIDs with defective Th17 immunity (e.g., autosomal dominant (AD) HIES, Caspase recruitment domain-containing protein 9 (CARD9) deficiency and AR Autoimmune polyendocrinopathy–candidiasis–ectodermal dystrophy (APECED) are examples of inborn errors of immunity associated with chronic mucocutaneous candidiasis (CMCC) (Table 3) [125,133].

PIDs associated with increased susceptibility to invasive fungal disease can be classified as phagocytic defects (Chronic granulomatous disease (CGD), Leukocyte adhesion deficiencies (LAD) and congenital neutropenia), cellular and combined immunodeficiencies (e.g., SCID, CIDs) and PIDs with defective Th17 immunity (AD-HIES, CARD9 deficiency). CGD and HIES are PIDs most frequently associated with IFDs (Table 4) [125,133].

A better understanding of the cellular and molecular mechanisms critical for anti-fungal immunity can be helpful for the development of new drugs and preventive measures, including vaccines [139].

## 4. Fungal Infections

The role of fungi as pathogens has been neglected since most are associated with opportunistic infections causing considerable morbidity and mortality. Recently, with the increase in the elderly population, improved treatments of immunosuppressed and transplanted patients, and premature neonates, the study of fungal diseases has gained special attention [140].

At the end of October 2022, the World Health Organization released its first Fungal Priority Pathogens List. In this List, fungal pathogens were divided into three categories: the critical priority group (*Cryptococcus neoformans*, *Candida auris*, *A. fumigatus*, and *Candida albicans*), the high priority group (*Candida glabrata*, *Histoplasma* spp., eumycetoma causative agents, Mucorales, *Fusarium* spp., *Candida tropicalis*, and *Candida parapsilosis*) and medium priority group (*Scedosporium* spp., *Lomentospora prolificans*, *Coccidioides* spp., *Candida krusei*, *Cryptococcus gattii*, *Talaromyces marneffei*, *P. jirovecii*, and *Paracoccidioides* spp.). According to this document, all stakeholders (e.g., policymakers, public health professionals, medical mycologists, pharmaceutics, and diagnostics technicians) should concentrate efforts to improve the overall response to fungal pathogens through strengthening laboratory capacity and surveillance, increasing the investments in research, development, and innovation, and improving epidemiology [141].

Worldwide estimations have revealed that nearly one billion patients are affected by superficial mycoses, 135 million suffer from mucosal infections, and several million present severe invasive infections frequently associated with extremely high mortality rates [142]. Moreover, invasive fungal infections are increasing yearly since new and rare pathogens are emerging and causing new diseases, such as *Kazachstania* spp. (also known as *Arxiozyma* spp.) [143], *Volvariella volvacea* [144], and *Rigidoporus cortisol* [145], among many others. All fungi-related diseases are responsible for approximately 1.6 million deaths yearly [142]. Although some of the fungal infections overlap between categories, for simplification, they may be divided into three distinct groups according to the infected tissue. The first group (Section 4.1) includes diseases that affect the skin and/or the mucosal surfaces; the second group (Section 4.2) includes those that affect subcutaneous tissues and the underlying tissues and organs; finally, the third group includes invasive and disseminated diseases (Section 4.3) (Appendix A).

### 4.1. Superficial Infections

Superficial fungal infections or mycoses involve a keratinized layer of skin, hair, nails and mucous membranes, caused by yeasts, dermatophytes and non-dermatophytes fungi.

Superficial mycosis of the skin, hair, and nails are among the most common mycoses worldwide, affecting nearly one billion people (25% of the population). At the same time incidence continues to increase yearly [146,147]. Socioeconomic conditions, geographic locations, and climate influence its prevalence and incidence [148,149].

#### 4.1.1. Dermatophytosis

Dermatophytes, the etiological agents of dermatophytosis, are keratinophilic fungi living on keratin-rich materials commonly found in the soil and skin (including hair and nails) [150,151]. This group includes the genera *Trichophyton*, *Microsporum*, *Epidermophyton*, *Lophophyton*, *Paraphyton*, *Nannizzia*, and *Arthroderma*. Most infections are caused by the three initial genera, which present different geographic distributions: *Trichophyton rubrum* presents a worldwide distribution, being significantly prevalent in developed countries from North America, Europe, Australia, and East Asia; *Microsporum canis* and *Trichophyton tonsurans* can be found in most European countries; *Trichophyton violaceum* is the most prevalent species in Middle Eastern countries, East Africa, and South China; *Microsporum audouinii* and *Trichophyton soudanense* are endemic to West African countries [152,153].

Several risk factors for dermatophytosis have been proposed. Among them are age (>60 years), gender (male patients), co-occurrence of other medical conditions (e.g., circulatory disorders, diabetes, excessive sweating, hypertension, ichthyosis, psoriasis, obesity, trauma, secondary immunodeficiencies), and race [154,155].

Transmission occurs by direct contact with an infected person or animal (cat, dog, guinea pig, or rabbit) or by contact with the soil. Transmission can also occur indirectly by transfer through wet surfaces (e.g., walkways, changing rooms, and foot washing stations in swimming pool facilities) or by sharing equipment (e.g., combs, hairbrushes, headrests, shaving brushes, and barber chairs, soaps, towels, bedding, and general articles of clothing and footwear, wrestling mats) [156].

From a clinical point of view, dermatophytosis or tinea is named according to the infection site. If the infection involves the foot, it is called *tinea pedis* or “athlete’s foot” (Figure 2A). in the scalp, is *tinea capitis* (Figure 2B); a rare variant is *tinea capitis favosa*, caused by *Trichophyton schoenleinii*, which occurs in children and adolescents; on the beard and moustache areas of men is *tinea barbae* (Figure 2C); in the hand region is *tinea manuum*; in the face is *tinea faciei*; on the nails and surrounding tissue is *tinea unguium*; in the upper body (trunk, shoulder, armpit, chest, and back) is *tinea corporis* (Figure 2D). Independently of the body location or etiological agent, when the infection is misdiagnosed and masked by topical corticosteroids, it is designated *tinea incognito* [151,157].

Due to the high number of etiological agents and body locations affected by dermatophytosis, myriad clinical manifestations can be highlighted. The leading dermatological sign of *tinea capitis* is alopecia, desquamation, and erythema. Other clinical types of cutaneous dermatophytosis present with erythema and desquamation. Pruritus is the primary symptom [154,158].

An early and accurate diagnosis is essential for choosing an efficient treatment that reduces transmission to other humans and improves outcomes. Since the occurrence of the etiological agents presents geographic trends, the first step is to perform a detailed medical and travel history. Secondly, a clinical examination of the patient, occasionally using dermoscopy, is essential.

Laboratory testing is frequently required to detect the presence of fungal elements, with direct microscopic examination of skin scrapings, nails, or hairs, using different stains (e.g., Calcofluor white) after clarification with potassium hydroxide. Histopathology is helpful in complex cases, particularly when deeper skin layers are affected. Active dermatophytosis includes parakeratosis, basket weave of the keratin layer, neutrophils in the base layers of the epidermis, spongiotic changes, eosinophils in the dermis, acanthosis or hyperkeratosis, and visualization of hyphae. Although isolating and identification of dermatophytes from a clinical sample grown in culture remains the “gold standard” for dermatophytosis diagnosis, several molecular techniques (e.g., DNA Polymerase Chain Reaction—PCR), protein by Matrix-assisted laser desorption ionization–time of flight mass spectrometry (MALDI–ToF MS) and antibody-based assays have been developed [151], promising a rapid and sensitive diagnosis.

Antifungal drug resistance of dermatophytes is alarming; 7 isolates out of 36 revealed resistances to one or more antifungal drugs (itraconazole, ketoconazole, and voriconazole associated with fluconazole). The two major classes used are azole, topical drugs (i.e., tioconazole, clotrimazole, oxiconazole, econazole, miconazole) and systemic azole agents (fluconazole, itraconazole) and allylamine (terbinafine and naftifine) [150]. Promising results are obtained with alternative antifungal agents such as epigallocatechin 3-O-gallate (EGCg) [159,160] and plant essential oils [161,162,163].

#### 4.1.2. Candidiasis

*Candida* species are typical resident commensal yeasts in healthy individuals’ oro-gastrointestinal and genitourinary tracts [164]. When the commensal balance in mucosae is disrupted in healthy individuals, *Candida* species locally overgrows on mucosal surfaces, resulting in candidiasis in a mucosal site (e.g., oral cavity, gastrointestinal tract, genitals, or other parts), causing moderate morbidity [165]. In an individual with secondary immunodeficiencies, the same yeast causes invasive infections that threaten mortality (see Section 4.3.2) [166]. Worldwide, more than 10 million patients suffer from mucosal candidiasis. The most common infections are oral candidiasis, vulvovaginitis, and balanitis (Figure 2E,F) [14].

Oral candidiasis, or “thrush”, is an opportunistic fungal infection affecting the oral mucosa [165]. Depending on the geographical locations, the main etiological agents are *C. albicans*, *C. glabrata*, *C. tropicalis*, *C. parapsilosis*, *C. krusei*, *C. dubliniensis*, and *C. guilliermondii* [167,168].

The principal risk factors include age (children younger than 5 years and adults older than 55 years), gender (females), use of medication (i.e., antibiotics, chemotherapy and radiotherapy medicines, corticosteroids), presence of underlying medical conditions (i.e., diabetes mellitus, hematologic malignancies such as leukemia, multiple myeloma, and lymphoma, hypothyroidism, iron deficiency, poor oral hygiene, wearing dentures, vaginal infections, and acquired immunodeficiency associated with leukopenia or HIV/AIDS) [169].

Transmission may occur by direct human-to-human contact or indirect contact through objects or surfaces, especially in immunocompromised patients.

Symptoms include ageusia, angular cheilitis, bleeding, denture stomatitis, edema, erythema, leukoplakia, paresthesia, pruritus, thrush, and xerostomia [169].

Vulvovaginal candidiasis (VVC) is also designated as vaginal thrush. This disease, which affects nearly 75% of healthy women during reproductive age, is due to abnormal yeast growth in the female genital tract mucosa [170,171].

Among the 20 *Candida* species known to cause VVC, the most common etiological agents are *C. albicans*, *C. glabrata*, *C. africana*, *C. dubliniensis*, *C. parapsilosis*, *C. tropicalis*, *C. krusei*, *C. guilliermondii*, and *C. lusitaniae* [171,172].

Risks or predisposing factors include the use of medication (i.e., antibiotics and glucocorticoids), and underlying medical conditions (e.g., diabetes mellitus, hormonal changes associated with puberty, pregnancy, hormone replacement therapy, obesity, secondary immunodeficiencies) [173]. Some behavioral aspects, such as using contraceptives, intrauterine devices, spermicides, condoms, insufficient sexual hygiene, and the use of tight-fitting pantyhose or wet clothes for long periods, favor the development of VVC [170].

The symptoms of VVC are nonspecific, being easily confused with other vaginal diseases, contributing to the misdiagnosis of this disease. The main signs and symptoms include edema, erythema, paresthesia, pruritus, odorless vaginal discharge with a cottage cheese appearance, vulvodynia, dyspareunia, and dysuria [170].

*Candida* balanitis is an inflammation of the glans penis that affects approximately 3–11% of males during their lifetime. Posthitis affects the foreskin (prepuce), and balanoposthitis includes the inflammation of the glans and the foreskin, occurring in nearly 6% of uncircumcised males. Since balanitis and balanoposthitis often co-occur, the terms are frequently applied synonymously [174].

Among the several etiological agents, *C. albicans* and *C. krusei* play a significant role in the disease’s outset. Such as other yeast infections, balanitis can be transmitted during vaginal, anal, and oral sex and through skin-to-skin contact [174].

Risk factors for balanitis include age (uncircumcised children under the age of 4 and middle-aged adults), the presence of foreskin, use of medications (e.g., topical and systemic corticosteroids and immunosuppressive drugs), underlying medical conditions (e.g., phimosis, diabetes mellitus, obesity, edematous conditions such as congestive heart failure and nephrosis, reactive arthritis, and sexually transmitted infections). Some behavioral aspects include unprotected sex, poor hygiene, nursing home environment, use of condom catheters, and sensitivity to chemical irritants (i.e., soaps and lubricants) [174].

Signs and symptoms include tight, shiny skin on the glans, redness around the glans, inflammation, soreness, itchiness, or irritation of the glans, a thick cheesy white discharge under the foreskin (smegma), an unpleasant smell, tight foreskin that cannot retract, painful urination, swollen glands near the penis, sores on the glans [174].

Candidiasis diagnosis includes a detailed medical history, symptom records, physical examinations, and laboratory tests. Direct microscopic examination and culture from oral exudate/genital secretions or glans swabs are the gold standard diagnostic tests [169,174,175].

Candidiasis treatments include topical and systemic azole agents (e.g., topical clotrimazole, ketoconazole or miconazole and systemic fluconazole and itraconazole) and topical nystatin. Alternative approaches for oral candidiasis include essential oils (e.g., *Allium tubeorosum*, *Cinnamomum cassia*, *Cinnamomum zeylanicum*, *Coriandrum sativum*, *Pelargonium graveolens*, and *Zataria multiflora*) and phytochemicals (citral, thymol, curcumin) [176,177,178,179].

In cases of recurrent vulvovaginitis or the presence of antifungal resistance strains, several alternative or complementary approaches have been developed, including the use of acid boric, flucytosine [180], ibrexafungerp [181], spilanthol [182], atorvastatin [183], probiotics [184], and p-Coumaric acid [178].

Antifungal agents are used with topical steroids (e.g., hydrocortisone) for severe balanoposthitis. In recurrent cases, circumcision is advised [174,185].

#### 4.1.3. Pityriasis Versicolor

*Pityriasis versicolor*, or *tinea versicolor*, is a common superficial mycosis worldwide. Most cases are reported in warm and humid conditions [186,187].

The disease is caused by a commensal yeast naturally occurring in the skin microflora. However, it grows out of control under specific conditions, forming small colonies. Although 14 species of *Malassezia* have been identified, only *Malassezia furfur*, *Malassezia globosa*, and *Malassezia sympodialis* have been isolated from skin lesions (Figure 2G,H) [187].

The risk factors for the development of pityriasis versicolor include oily skin, hormonal changes due to puberty and pregnancy, immunodeficiencies, behavioral aspects (application of oily lotions and creams, excessive sweating), and environmental conditions (warm and humid weather) [187,188].

The most common signs and symptoms include hyperpigmented or hypopigmented macules, pruritus, and scaling, mainly affecting body locations such as the trunk, neck, and proximal extremities [187].

The diagnosis is usually based on clinical examination. Lab tests are performed for epidemiologic studies or complex clinical cases (microscopic examination from skin scrapings of the infected areas and culture) [187].

Treatment includes the use of topical medication (azole agents, pyrithione zinc, selenium sulfide, sulfur associated with salicylic acid) and, occasionally, oral antifungal medication (itraconazole, fluconazole) [187].

### 4.2. Subcutaneous Infections

#### 4.2.1. Eumycetoma

Mycetoma, an infection of the skin and underlying structures, is mainly identified in tropical and sub-tropical areas (e.g., India, Mexico, Venezuela, Chad, Ethiopia, India, Mauritania, Mexico, Senegal, Somalia, Sudan, and Yemen). This disease is associated with poor and socioeconomically impaired communities [189,190].

Caused by more than 30 fungal species, it is characterized by lengthy disease duration and low recovery rate. Case management relies on an accurate diagnosis by adequately identifying the causative agent and evaluating its susceptibility to the available drugs [191]. The most common etiological agents of eumycetoma include *Madurella mycetomatis* (>70% of the cases), *Microsporum grisea*, *Pseudallescheria boydii*, *Leptosphaeria senegalensis*, *C. lunata*, *Scedosporium apiospermum*, *Neotestudina rosatii*, *Acremonium* spp., and *Fusarium* spp. [192].

The risk factors include age (20–40 years), gender (male), underlying medical conditions (e.g., secondary immunodeficiencies), and environmental exposure [192].

Transmission occurs when fungal spores in soil and water enter the body through wounds or other minor skin injuries caused by thorn pricks, wood splinters, or accidental cuts. No person-to-person transmission has been reported [192].

The infection manifests as painless nodules under the skin at the injury site. The small initial mass grows over time due to the accumulation of large aggregates of fungal hyphae and neutrophils, and the release of cytokines and enzymes that digest subcutaneous tissues and create multiple granulomas and abscesses. Abscesses drain with the extrusion of granules or grains. The infected limb is deformed in untreated or poorly treated cases, and the underlying bone and muscular tissues are destroyed (Figure 2I–L) [192].

Diagnosis relies on medical and travel history, symptom records, physical examinations, lab tests (direct microscopic examination, histopathological examination, culture growth, molecular profiling, and serodiagnosis test), and imaging tests (X-rays, ultrasound, and MRI) [192].

Eumycetoma treatment includes the use of oral antifungal medication (i.e., itraconazole, ketoconazole, lanoconazole, luliconazole, posaconazole, ravuconazole, voriconazole). Nevertheless, more complex and incontrollable cases may require surgery and amputation [191,192,193]. Several phytotherapeutic alternatives, such as *Acacia nubica*, *Nigella sativa*, *Boswellia papyrifera*, *Melaleuca alternifolia*, *Moringa oleifera*, *Zingiber officinalis*, *Piper nigrum*, *Eugenia caryophillus*, and *Cinnamomum verum*, have been suggested, with positive effects on the treatment of this disease [194].

#### 4.2.2. Chromoblastomycosis

Like Eumycetoma, chromoblastomycosis is endemic in tropical and sub-tropical regions between 30° North and 30° South [195].

The most common etiological agents are *Fonsecaea pedrosoi*, *F. monophora*, *F. nubica*, *F. pugnacius*, *Cladophialophora carrionii*, *C. yegresii*, *C. samoensis*, *Phialophora verrucosa*, *Rhinocladiella aquaspersa*, *R. tropicalis*, *R. similis*, and *Cyphellophora ludoviensis*. As common characteristics, all these species present low pathogenicity and are saprophytes from the soil, plants, and decomposing organic matter [195,196]. Most of these fungi are pigmented or dematiaceous fungi.

Risk factors include age (50–60 years), gender (male), underlying medical conditions (secondary immunodeficiencies and poor nutrition), and environmental exposure (e.g., lack of protective shoes, gloves, or garments, and hygienic habits). This disease is considered an occupational disease, highly prevalent among farm laborers, loggers, and vendors of farm products in endemic regions.

Transmission occurs when the fungal spores in the soil, plants, and decomposing matter, enter the human body (mainly lower members) through wounds or other minor skin injuries.

Chromoblastomycosis presents with the formation of nodules or papules with centrifugal growth—moriform bodies—at the infection site. The initial lesion remains stable for several months or years and evolves into metastatic lesions (nodular, verrucous, plaque, and tumoral) at other anatomic sites [195].

The diagnosis is based on medical and travel history, symptom records, physical examinations, and lab tests (direct microscopy, histopathology, culture from biopsy, intradermal tests, and molecular biology) [195,197].

The treatment includes oral antifungal medication (itraconazole, terbinafine, voriconazole, ravuconazole, posaconazole, isavocunazole, fluconazole, amphotericin B, imiquimod), surgery (to remove lesions), cryosurgery (liquid nitrogen), thermotherapy (42–46 °C to inhibit fungal growth), CO_2_ laser, and photodynamic therapy [195,197].

### 4.3. Systemic Infections

#### 4.3.1. Aspergilloma and Aspergillosis

Conidia from *Aspergillus* species are an ever-present component of the airborne microflora from indoor and outdoor environments continuously inhaled by humans. These conidia present reduced dimensions (2.5 to 3.5 μm in diameter). They are coated by a hydrophobic layer that helps them to evade the filtration in the nose and enter the respiratory airways. In healthy individuals, conidia are effectively trapped by the mucociliary clearance mechanisms, inactivated, and eliminated. However, if conidia reach the alveoli, several immunological cellular defenses are activated to clear these particles. In immunodeficient patients, the conidia can evade this defense mechanism clearance from the respiratory tract. As such, conidia that reach deeper regions of the respiratory tract germinate and establish fungal masses inside the lungs [198].

Aspergillosis is a group of illnesses dependent on host factors and their immunologic response. Noninvasive forms include allergic bronchopulmonary aspergillosis (Section 2.1) and allergic fungal rhinosinusitis (Section 2.4). Invasive forms include chronic pulmonary aspergillosis (i.e., aspergilloma) and invasive pulmonary aspergillosis.

The principal etiological agent associated with aspergillosis is *A. fumigatus*. However, other non-*fumigatus Aspergillus* species also contribute to the disease outset; among these are: *A. niger*, *A. terreus*, *A. nidulans*, and *A. oryzae* (Appendix A) [198,199].

Aspergillosis risk factors include the use of medications (antibacterial agents, corticosteroids, anti-tuberculosis drugs, immunosuppressants), the existence of underlying medical conditions (chronic obstructive pulmonary disease, severe viral pneumonia, severe influenza, severe COVID, tuberculosis, sarcoidosis, pneumoconiosis, lung cancer, liver failure, liver cirrhosis, neutropenia, hematological malignancies, transplantation, HIV/AIDS, and other immunodeficiency diseases) [198,200,201]. Some additional factors may also play a role in the disease onset, such as malnutrition, marijuana smoking, and a stay in a hospital setting (nosocomial infection) [199,202].

The main route of infection is through the respiratory tract by spore inhalation. Nonetheless, *Aspergillus* conidia can enter the human body through other tissues, such as the skin, central nervous system, eyes, and nails, disseminating throughout the body [202]. No person-to-person transmission has been reported.

An aspergilloma is a fungus ball formed by a mixture of *Aspergillus* hyphae, cellular debris, and mucus. Since the fungus colonizes a preexisting cavity in the lung parenchyma, it is considered a noninvasive form of chronic pulmonary aspergillosis. The global burden of aspergilloma is nearly 1.2 million, corresponding to the higher incidence and prevalence rates in lower-income countries in Africa, the western Pacific, and Southeast Asia. Although fungal balls mainly develop in the lung, they can also be found in other organs, such as nasal sinuses, the heart, and the brain [199].

Invasive pulmonary aspergillosis (IPA) is a severe infection with rapid dissemination from the lungs to the brain, heart, kidneys, or skin [202]. It is the primary cause of death in immunocompromised patients (e.g., patients with leukemia and recipients of hematopoietic stem cell transplants), corresponding to a mortality rate between 70 and 90% [203].

The main clinical manifestations include arthralgia, cephalalgia, chills and fever, chest pain, dry cough, cough with hemoptysis, wheezing, dyspnea, eye symptoms, skin lesions, fatigue, and weight loss [199,202].

Early diagnosis is essential for a favorable prognosis. It includes a detailed medical history, symptoms description, physical examinations, lab tests (direct microscopy, histopathology, detection of *Aspergillus* antigen, detection of IgG and IgM antibodies against *Aspergillus*, culture, and molecular identification from biopsy, blood, sputum, tracheal aspirate, and broncho-alveolar lavage), and imaging tests (X-ray, CAT scan with contrast) [204].

#### 4.3.2. Candidemia and Invasive Candidiasis

Candidiasis, a general medical term, includes cutaneous and mucosal infections (Section 4.1.2) and invasive candidiasis (Appendix A). Invasive candidiasis includes candidemia (bloodstream infections with *Candida* spp.) and deep-seated infection (intra-abdominal abscess, peritonitis, and osteomyelitis) [205,206], which can occur either simultaneously or independently.

More than 15 *Candida* spp. cause invasive candidiasis. Among them, *C. albicans* is the most common etiological agent. However, an increasing number of infections can be attributed to non-albicans *Candida* species, such as *C. glabrata*, *C. auris*, *C. parapsilosis*, *C. tropicalis*, and *C. krusei* [207]. Nevertheless, each species’ prevalence varies according to geographical location [205].

Candidemia is associated with the translocation of commensal *Candida* from the gastrointestinal tract to the bloodstream or contamination/colonization of an intravenous catheter. Deep-seated candidiasis results from the non-hematogenous introduction of *Candida* species into sterile sites, mainly the abdominal cavity, following gastrointestinal tract disruption or via an infected peritoneal catheter [208]. Invasive candidiasis is frequently associated with healthcare settings and is the fifth most significant infection in healthcare worldwide. It presents high morbidity, mortality, and costs [209]. Risk factors for the development of invasive candidiasis include extended stays at intensive care units, use of medication (antibiotics, cancer chemotherapy, corticosteroids), underlying medical conditions (premature neonates, diabetes mellitus and other secondary immunodeficiencies), major abdominal surgery, use of implanted medical devices (vascular catheters, parenteral feeding catheters, and prosthetic heart valves) [207].

Invasive candidiasis does not present specific clinical signs or symptoms, including abdominal pain, cephalalgia, chills, fatigue, fever, light sensitivity, low blood pressure, memory loss, mental confusion, myalgia, skin rash, vision changes, and weakness [205].

The early diagnosis of invasive candidiasis is challenging and essential to effectively managing the disease [205]. The diagnostic steps include elaborating a detailed medical history, symptoms records, physical examinations, and lab tests (direct microscopy, detection of *Candida* antigens and anti-*Candida* antibodies, molecular identification, culture from biopsy and blood) [208,210].

#### 4.3.3. Cryptococcosis

Cryptococcosis, an invasive disease caused by the yeastlike fungus *Cryptococcus* spp., has become a significant infection in both immunocompromised and immunocompetent hosts, frequently associated with significant mortality rates (70% in low-income; 20–30% in high-income countries) [211,212].

The etiological agents include members of the *Cryptococcus gattii*/*Cryptococcus neoformans* species complexes (Appendix A) [212,213]. The two species occupy distinct ecological niches. *Cryptococcus neoformans*, mainly isolated from the soil and bird excrements [214], present worldwide distribution, infecting immunosuppressed individuals (mainly with HIV/AIDS) [215]. *Cryptococcus gattii*, mainly recovered from decaying wood and other plant materials [214], causes 70% to 80% of infections in immunocompetent hosts [215].

Infection occurs by inhaling desiccated yeast cells or basidiospores into the lungs. These particles may spread to several organs, assuming various clinical forms (e.g., pulmonary, meningitis, meningoencephalitis, visceral, osseous, mucocutaneous, and cutaneous). No person-to-person transmission has been reported [214]. The primary risk factors for the disease include age (40–60 years), use of medication (chemotherapy, corticosteroids, immunosuppressors), and underlying medical conditions (lung diseases, diabetes mellitus, malignancy, HIV/AIDS, transplantation). A behavioral aspect associated with predisposition factors for this disease is tobacco smoking [216,217].

In cases of meningitis, symptoms include cephalalgia, cervicalgia, mental confusion, nausea and vomiting, and sensitivity to light. In the case of pulmonary disease, symptoms include fever, chills, cough, malaise, sleep hyperhidrosis, dyspnea, weight loss, and hemoptysis [211].

Diagnosis is achieved through medical history, symptoms records, physical examinations, lab tests (direct microscopy, histopathology, culture, antigen detection, and molecular identification from biopsy, serum, cerebrospinal fluid, broncho-alveolar lavage, and urine), and imaging tests (X-ray, CT scan, and MRI) [211,215,218].

#### 4.3.4. Blastomycosis

Blastomycosis is a fungal infection caused by *Blastomyces* spp. (e.g., *B. dermatitis*, *B. gilchristii*, *B. helices*, *B. parvus*, and *B. percursus*)—a dimorphic pathogen (Appendix A). Blastomycosis is mainly endemic to the southeastern United States and Canada, where the annual incidence is less than one case per 100,000 individuals [219,220]. In North America, the area of endemicity includes the southeastern and south-central states, especially those bordering the Ohio and Mississippi River basins, the Midwest states and Canadian provinces bordering the Great Lakes, and an area in New York and Canada along St. Lawrence Rivera. Blastomycosis is also endemic in Africa [221,222]. While the mortality rate ranges between 4 and 22%, this depends on patient characteristics [223].

The fungus lives in soils with dense vegetation near rivers or other water [220,223]. Spores became aerosolized upon colony disturbance associated with humans (e.g., fishing, hunting, construction) and animal activities (e.g., pets) [220]. As such, fungal particles can be inhaled, reaching the lower respiratory tract. Conidia are phagocytized by bronchopulmonary mononuclear cells and killed by neutrophils and macrophages, and the patient remains asymptomatic. However, if Blastomyces is in its yeast-like form, the thick wall avoids phagocytosis and killing, and the patient develops a symptomatic infection [220]. The lungs are the most affected organs. However, other body sites can also be involved (e.g., skin, bones, the genitourinary system, and the central nervous system), and symptoms resemble those from other infections and/or malignant diseases [220,223].

Transmission occurs by environmental exposure through inhaling spores frequently found in soils from wooded areas. No person-to-person transmission has been reported [219]. The risk factors associated with blastomycosis include age (adult), gender (male), and underlying medical conditions (e.g., pulmonary multilobar disease, obesity, diabetes mellitus, and immunosuppression) [220,224].

The primary symptoms include chest pain, chills, fever, myalgia, sleep hyperhidrosis, dyspnea, cough (productive or non-productive), hemoptysis, fatigue, anorexia, and weight loss [223].

As with most endemic mycoses, symptom severity and disease course depend on exposure extent (inoculum size) and the immune status of the exposed individual. Based mainly on blastomycosis outbreak studies, the symptomatic disease occurs in less than half of infected individuals. Clinical illness caused by *B. dermatitidis* may present as pulmonary or extrapulmonary disseminated disease.

A classic form of blastomycosis is chronic cutaneous involvement that is almost always the result of hematogenous dissemination from the lung, in most instances without evident pulmonary lesions or systemic symptoms. Pulmonary blastomycosis may be asymptomatic or present as a mild flulike illness. More severe infection resembles bacterial pneumonia with acute onset, high fever, lobar infiltrates, and cough. Progression to fulminant adult respiratory distress syndrome with high fever, diffuse infiltrates, and respiratory failure may occur. A more subacute or chronic respiratory form of blastomycosis may resemble tuberculosis or lung cancer, with the radiographic presentation of pulmonary mass lesions or fibronodular infiltrates.

Clinical suspicion is essential for diagnosis, and delays can be avoided based on laboratory testing.

Diagnosis requires a detailed medical and travel history, symptoms record, physical examinations, lab tests (histopathology, culture, antibody testing, *B. dermatitidis* antigen detection, and molecular identification from tracheal aspirates, sputum, bronchoalveolar lavage, bone, blood, cerebrospinal fluid (CSF), prostatic tissue, urine), and imagological tests (X-ray) [224,225].

#### 4.3.5. Coccidioidomycosis

Coccidioidomycosis (also known as Valley Fever), caused by *Coccidioides immitis* and *C. posadasii* from soils, is a group of fungal infections of the Western hemisphere and is endemic to areas with reduced rainfall levels, occurring between the latitudes 40° north and 40° south [226,227,228] (Appendix A). The disease ranges from mild, self-limited, febrile illnesses to severe life-threatening infections [229].

The transmission route is established by inhaling arthroconidia that become airborne due to natural (e.g., earthquakes, dust storms, fires) and anthropogenic events (e.g., military maneuvers, recreational activities, agriculture, construction). Although no person-to-person transmission has been established, sporadic cases of transmission by transplanted organs (attributed to infected donor lungs, liver, and kidneys) and neonatal transmission (due to aspiration of infectious vaginal secretions during birth) have been reported [227]. The most relevant risk factors for disease onset include age (>80 years), race (non-Caucasian races), use of medication (e.g., corticosteroids, anti-tumor necrosis factor medications, chemotherapy, other immunomodulation therapies), underlying medical conditions (HIV/AIDS, pregnancy, diabetes mellitus, transplantation), and environmental exposure [227,229].

Most of patients (60%) with coccidioidomycosis remain asymptomatic. One-third of the patients develop a pulmonary illness; in 0.5–2% of cases, the infection disseminates to other body locations (e.g., skin, bones, joints, central nervous system). The most common symptoms include arthralgia, cephalalgia, chest pain, chills, cough, hemoptysis, dyspnea, fatigue, fever, myalgia, lung nodules, rashes, sleep hyperhidrosis, and weight loss [227].

Diagnosis includes detailed medical history, symptoms records, physical examinations, and lab tests (direct microscopy, histopathology, detection of *Coccidioides* antigens, molecular identification, culture from biopsy and blood) [228,229].

#### 4.3.6. Histoplasmosis

Histoplasmosis, caused by *Histoplasma capsulatum*, is a global disease endemic to regions distributed through the six inhabited continents (Appendix A). The current climate change and anthropogenic land use are changing the conditions suitable for the development of the fungus, resulting in changes in its epidemiology [230].

Infection occurs by inhaling aerosolized microconidia from nitrogen/phosphate-enriched soils contaminated with bird or bat droppings [231]. There is no person-to-person transmission, but sporadic transmission cases by transplanted organs have been reported [232]. The primary risk factors include age (<5 years; >55 years), use of medication (chemotherapy medicines, corticosteroids, immunosuppressors, tumor-necrosis-factor inhibitors), and underlying medical conditions (lung diseases, transplanted patients, HIV/AIDS and other secondary immunodeficiencies) [231].

Histoplasmosis symptoms resemble other diseases, such as community-acquired pneumonia, tuberculosis, sarcoidosis, Crohn’s disease, and malignancy. The symptoms include arthralgia, cephalalgia, chest pain, chills, cough, fatigue, fever, and myalgia [230].

Accurate diagnosis is challenging and limited in countries where histoplasmosis is highly endemic. Diagnostic steps include a medical history, symptoms, physical examinations, lab tests (direct microscopy, histopathology, culture, detection of specific antibodies to *H. capsulatum*, antigen detection, molecular identification from urine, serum, bronchoalveolar lavage, and cerebrospinal fluid), and imaging tests (X-ray or CT scan) [231,233].

Histoplasmosis can be cured without medication in most patients. However, treatment is recommended in immunocompromised patients and those with progressive disseminated or acute pulmonary disease [234].

#### 4.3.7. Emergomycosis

Emergomycosis, a recently emerged systemic fungal infection, is caused by the novel dimorphic fungus species *Emergomyces* [235]. The causative agents include *E. pasteurianus*, *E. africanus*, *E. canadensis*, *E. orientalis*, and *E. europaeus* [235,236,237] (Appendix A).

Infection occurs through inhaling airborne conidia released from saprophytic mycelia in soil. In the human body, conidia are converted into yeast-like cells that replicate and disseminate to other body parts (e.g., skin, lungs, liver, spleen, bone marrow, lymph nodes, brain, and cervix) [235]. Risk factors include the use of medication (immunosuppressors) and underlying health conditions (e.g., HIV/AIDS, neutropenia, solid organ transplantation, hematological malignancies, and diabetes mellitus) [235].

Cutaneous manifestations include umbilicated papules, nodules, ulcers, verrucous lesions, crusted hyperkeratotic plaques, and erythema. Respiratory manifestations include epistaxis, nasal congestion, oroantral fistula, pneumonia, and lobar atelectasis. Hematological manifestations include anemia and thrombocytopenia. Central nervous system manifestations include changes in mental status, headache, seizure, ataxia, vision changes, and behavioral changes. Gastrointestinal manifestations are associated with changes in different enzymes. Genital manifestations include the appearance of endocervical masses [235].

Diagnosis relies on medical history, symptoms records, physical examinations, and lab tests (histopathology, culture, and molecular identification from blood, skin tissue, bone marrow aspirate, trephine biopsy, lymph node aspirate, sputum, and bronchoalveolar lavage) [235].

So far, no treatment guidelines are available. Therefore, current therapeutic approaches are based on those applied to similar disseminated fungal diseases [237].

#### 4.3.8. Paracoccidioidomycosis

Paracoccidioidomycosis (also known as South American blastomycosis), an endemic fungal disease from Latin American countries, presents a greater prevalence in Central and South America [238,239]. The etiological agents are *Paracoccidioides americana*, *P. restrepiensis*, *P. venezuelensis*, and *P. lutzii* [238,240,241] (Appendix A).

Transmission routes include the inhalation of airborne fungal conidia or the entry of spores through minor traumas. No person-to-person transmission [242]. The risk factors associated with disease onset are age (>30–60 years), gender (male), race (Africans), and underlying medical conditions (secondary immunodeficiencies, malnutrition, poor hygiene). Additionally, behavioral aspects, such as alcohol and tobacco consumption, can play a significant role in disease development [239,242].

Paracoccidioidomycosis symptoms resemble numerous other infections (e.g., tuberculosis) and noninfectious diseases (e.g., sarcoidosis) and include cough, dyspnea, fatigue, fever, hepatosplenomegaly, lesions in the mouth and throat, lymphadenitis, weight loss [239].

Diagnosis includes elaborating a medical and travel history, symptom reports, physical examinations, lab tests (direct microscopy, histopathology, and culture from biopsy samples of the affected body part), and imaging tests (MRI) [241,243].

#### 4.3.9. Talaromycosis

Talaromycosis (previously designated as penicilliosis), is a life-threatening mycosis caused by the fungus *T. marneffei* (Appendix A). This disease is frequently considered the leading cause of death in patients with advanced HIV and other immunosuppressive conditions from South and Southeast Asia [244,245]. More than 17,000 talaromycosis cases and nearly 5000 associated deaths have been reported [246]. In severely immunocompromised patients, the disease is disseminated involving the lungs, liver, spleen, gastrointestinal tract, bloodstream, skin, and bone marrow [246].

The main transmission route of infection occurs by spore inhalation. Another route is established when fungal spores live in the soil and plants and enter the body through wounds or other minor skin injuries [246]. The primary risk factors include underlying medical conditions (HIV/AIDS, autoimmune diseases, hematological malignancy, solid organ and bone marrow transplantations, chronic obstructive pulmonary disease, lung malignancy, tuberculosis, and sarcoidosis) [247].

The symptoms of infection are agitation, anemia, confusion, depressed consciousness, dyspnea, fatigue, fever, hepatosplenomegaly, lymphadenopathy, respiratory and gastrointestinal abnormalities, skin lesions (papules with central necrosis, predominantly on the head and upper chest; it can also occur in the form of pustules, nodules, subcutaneous abscesses, cysts, or ulcers), and weight loss [244].

Diagnosis includes a medical and travel history, symptoms records, physical examinations, and lab tests (direct microscopy, histological analysis, antigen detection, molecular identification, and culture from blood and biopsies of smear skin, lymph nodes, and bone marrow aspirates) [244].

#### 4.3.10. Fusariosis

Fusariosis, caused by *Fusarium* spp., is a disseminated fungal disease (Appendix A). Fungus belonging to this genus can be ubiquitously found in the environment (soil, water, air, and plants). It is associated with producing toxins responsible for food poisoning cases. In humans, depending on the predisposing factors and the host immunological status, *Fusarium* spp. cause localized (e.g., onychomycosis, skin infections, and keratitis) or disseminated diseases (i.e., fusariosis) [248].

*Fusarium* species present a worldwide distribution, with several species complexes associated with disease onset. Among the most relevant are *F. solani* (40–60% of cases), *F. oxysporum* (nearly 20% of cases), *F. fujikuroi*, *F. moniliforme*, *F. incarnatum-equiseti*, *F. clamydosporum*, *F. dimerum*, *F. sambucinum*, *F. concolor*, and *F. lateritium* [249,250].

Transmission occurs when fungal spores in the soil and plants enter the body through wounds or other minor skin injuries. The main risk factors are secondary immunodeficiencies (neutropenia, hematologic malignancies) [248,249].

Symptoms inlcude fever, myalgia, pneumonia, skin lesions, superficial infection in the feet with lymphangitis, and toxaemic appearance [250].

Diagnosis includes the medical history, symptoms record, physical examinations, and lab tests (histopathology, culture, and molecular identification from blood) [251,252].

#### 4.3.11. Hyalohyphomycosis

Hyalohyphomycosis includes several diseases caused by opportunistic hyaline non-dematiaceous molds or yeasts, such as *Aspergillus* spp., *Acremonium* spp., Basidiomycota, *Beauvaria* spp., *Chaetoconidium* spp., *Chrysosporium* spp., *Fusarium* spp., *Microascus* spp., *Paecilomyces* spp., *Penicillium* spp., *Pseudallescheria* spp., *Scedosporium* spp., *Schizophyllum* spp., *Scopulariopsis* spp., and *Trichoderma* spp. (Appendix A). These fungi cause superficial or localized infections in immunocompetent patients and invasive or disseminated infections in immunocompromised hosts [253,254].

Routes of infection include inhaling aerosolized spores from the soil, polluted waters, and decaying organic materials, entering fungal spores through wounds or other minor skin injuries, and ingesting contaminated food products. Among the reported risk factors are the use of medication (immunosuppressors), underlying medical conditions (HIV/AIDS, autoimmune disorders, transplanted patients, neutropenia, malignancy, liver diseases, kidney diseases), trauma, burns, surgery, and the use of a central venous catheter [253,254].

The most significant hyalohyphomycosis symptoms include anemia, arthralgia, cellulitis, encephalitis, mental confusion, cough, endophthalmitis, fever, hepatosplenomegaly, keratitis, lymphadenopathy, onychomycosis, osteomyelitis, peripheral edema, peritonitis, pneumonia, renal failure, sinusitis, and weight loss [252,254].

Diagnosis is performed by elaborating a medical history, symptoms record, physical examinations, and lab tests (histopathology immunohistological staining, culture, and molecular identification from biopsy, blood, and skin) [253,254].

#### 4.3.12. Lomentosporiosis

Lomentosporiosis is an emerging fungal invasive disease caused by *L. prolificans* (previously known as *Scedosporium prolificans*) (Appendix A) (Figure 2I,J). Depending on the immunological status of the individual, this fungus induces a broad spectrum of clinical manifestations that range from superficial (in immunocompetent individuals) to disseminated infections (in immunocompromised patients) [255,256].

This fungus has been isolated from several environmental sources (e.g., oil-soaked soils, cattle dung, sewage, polluted waters, plants, chicken manure, and other animals) from different locations with dry climates (e.g., Australia, USA, and Spain). Two main routes of entry in the human have been described: inhalation of airborne conidia and traumatic inoculation of conidial cells from contaminated environmental sources. Risk factors for lomentosporiosis development include medication (corticosteroids, immunosuppressive therapy), underlying medical conditions (solid organ transplant, hematopoietic stem cell transplant, acute leukemia, neutropenia, acquired immunodeficiency syndrome, cystic fibrosis, cavitary lung disease, endocarditis), surgery, and trauma. Behavioral aspects such as intravenous drug use can facilitate disease development [256].

Cutaneous infection signs include numerous erythematous nonpruritic skin nodules with or without a necrotic center. Respiratory infection symptoms are cough, dyspnea, fever, and pleuritic chest pain. Cardiac infection presents with fever and embolic phenomena. Central nervous system involvement includes meningitis, meningoencephalitis, and brain abscess formation. Eye infection includes endophthalmitis, kerato-scleritis, and kerato-uveitis. Moreover, other clinical manifestations, such as mycotic aneurysms, external otitis, sinusitis, peritonitis, onychomycosis, and esophagitis, have been previously linked to lomentosporiosis [255,256].

Diagnosis includes elaborating a medical history, symptoms record, physical examinations, and lab tests (direct microscopy, histopathology, culture, and molecular identification) [256,257].

#### 4.3.13. Mucormycosis

Mucormycosis (previously known as zygomycosis) is an angio-invasive fungal infection initiated by the fungi belonging to the order Mucorales. This disease is associated with high morbidity and mortality rates. The species most frequently isolated from patients are *Apophysomyces* (*A. variabilis*), *Cunninghamella* (*C. bertholletiae*), *Lichtheimia* (*L. corymbifera*, *L. raosa*), *Mucor* (*M. circinelloides*), *Rhizopus* (*R. arrhizus*, *R. microsporus*), *Rhizomucor* (*R. pusillus*), and *Saksenaea* (*S. vasiformis*) [258,259] (Appendix A).

Humans develop mucormycosis primarily by inhalation of sporangiospores, and occasionally by ingesting contaminated food or traumatic inoculation. No person-to-person transmission has been reported. Risk factors for disease development include the use of medication (corticosteroid, immunosuppressors, and chemotherapy medicines), underlying medical conditions (HIV/AIDS, neutropenia, malignancy, transplanted patients, diabetes mellitus, hemochromatosis, prematurity and low birth weight, skin injury due to surgery, burns, or wounds). Behavioral aspects include injectable drugs and chronic alcoholism [258].

Mucormycosis mild infections involve the cutaneous tissue. More severe and disseminated forms can affect several organs, such as the brain, lungs, gastrointestinal tract, kidney, bones, heart, ear, parotid gland, uterus, urinary bladder, and lymph nodes. Therefore, a wide range of signs and symptoms can be associated with this disease. Among them are abdominal pain, cephalalgia, chest pain, mental status changes, coma, cough, dyspnea, fever, nausea and vomiting, gastrointestinal bleeding, lesions on the nasal bridge or palate, nasal or sinus congestion, and unilateral facial edema [258].

Diagnosis comprises medical history, symptom record, physical examinations, lab tests (direct microscopy, histopathology, culture, antigen detection, and molecular identification from blood, respiratory fluid, and biopsy), and imaging tests (CT scan) [260].

#### 4.3.14. Phaeohyphomycosis

Phaeohyphomycosis includes several diseases caused by opportunistic “dematiaceous” or “melanized” fungi, distinguished by the presence of melanin in their cell walls, considered a virulence factor in these organisms. These fungi belong to the genera *Alternaria*, *Aureobasidium*, *Biatriospora*, *Bipolaris*, *Chaetomium*, *Cladophialophora*, *Corynespora*, *Curvularia*, *Exophiala*, *Exserohilum*, *Fonsecaea*, *Microsphaeropsis*, *Phialemonium*, and *Ramichloridium* [261,262] (Appendix A) (Figure 2K,L).

Dematiaceous fungi are usually found in the soil, growing on plants, and in organic debris from worldwide locations. Humans acquire these microorganisms by inhalation of aerosolized spores or by immersion in contaminated freshwater [261]. The primary risk factors for developing this disease include medication (chemotherapy, corticosteroids, immunosuppressors) and underlying medical conditions (cancer, chronic sinusitis, neutropenia, HIV/AIDS). Behavioral aspects, such as intravenous drugs, may influence disease onset [261,262].

These microorganisms are responsible for the onset of several invasive manifestations, such as deep local infections, pulmonary infection, cerebral infection, and disseminated disease, associated with high mortality rates. Therefore, the primary symptoms include behavioral changes, cough, dyspnea, fever, gastrointestinal bleeding, mental confusion, nausea and vomiting, seizures, sepsis, skin rashes, and ulcers [261].

Diagnosis includes elaborating a detailed medical history, symptoms record, physical examinations, and lab tests (histopathology and culture) [261].

#### 4.3.15. Pneumocystis Jirovecii Pneumonia

*Pneumocystis* pneumonia (PCP) (also referred to as *P. jirovecii* pneumonia—PJP—or pneumocystis), is caused by *P. jirovecii* (Appendix A). In immunocompetent individuals, this infection is asymptomatic or only presents mild respiratory symptoms. In immunocompromised patients, it is a severe disease associated with high morbidity and mortality rates [263,264]. Moreover, PCP is a common healthcare-associated infection, being person-to-person transmission made through the air, mainly by droplets [265]. Nearly 400,000 cases are reported annually [265].

Initially reported in premature infants and malnourished children after the end of World War II, this disease was later observed in other patient populations. Among the most affected are patients that use medication (corticosteroid, chemotherapy medication, immunosuppressors, antiretroviral therapy) and with underlying medical conditions (pulmonary diseases, inflammatory diseases, autoimmune diseases, HIV/AIDS, solid organ transplant, malignancies, rheumatic diseases, hematologic diseases) [263,265].

As with many other fungal diseases, PCP symptoms are nonspecific and include chest pain, chills, nonproductive cough, dyspnea, hypoxemia, respiratory failure, fatigue, and fever. The lungs are the primary site of infection. Nevertheless, rare cases of extrapulmonary PCP have been reported in humans, including the eyes, ears, lymph nodes, liver, spleen, and bone marrow. Systemic dissemination has also been documented [263].

The diagnostic steps include elaborating a medical history, symptoms record, physical examinations, lab tests (direct microscopy, histopathology, antigen detection, and molecular identification from serum, sputum, BAL, bronchial washing fluid, and lung biopsy), and imageology tests (X-ray) [263,266,267].

#### 4.3.16. Sporotrichosis

Sporotrichosis is a fungal disease caused by the *Sporothrix* spp. (Appendix A). This disease is prevalent in tropical and subtropical areas, with endemic areas in America, Africa, and Asia [268,269]. More frequently presented as a subcutaneous mycosis, other body parts (e.g., mucosa, bones, joints, central nervous system) or even systemic and disseminated diseases can be observed, depending on the inoculum size, infection route, strain virulence, and the host immunological status [270].

The etiological agents are *S. schenckii*, *S. brasiliensis*, *S. innfecti*, *S. luriei*, and *S. pallida*. Species belonging to this genus are saprophytes living on organic matter, dead wood, mosses, hay, and cornstalks [268].

The infection route includes contact with contaminated plant material followed by traumatic inoculation of fungal propagules into skin tissue (sapronosis). Although rare, infection by inhalation of aerosolized fungal propagules has also been reported. Over the last few years, the epidemic of sporotrichosis outbreaks in domestic animals (zoonosis) in Asia and South America demonstrated that the disease could also be disseminated when humans have close contact with animals (e.g., cat-to-human or armadillo-to-human). No person-to-person transmission has been reported [269,270,271]. Risk factors for disseminated sporotrichosis include age (middle age), gender (male), the use of medication (corticosteroids), and the presence of underlying medical conditions (diabetes mellitus, chronic obstructive pulmonary disease, and secondary immunodeficiencies). Behavioral aspects such as alcohol consumption, rose gardening, farming, mining, horticulture, and armadillo hunting can increase the exposure risk [272].

Symptoms include arthralgia, cephalalgia, chest pain, cough, dyspnea, fever, mental confusion, seizures, skin nodules, and weight loss [272].

Diagnosis includes the elaboration of a detailed medical and travel history, symptoms record, physical examinations, lab tests (direct microscopic, cytopathology, histopathology, serology, culture, molecular identification from biopsy and blood), and imagological tests (X-ray) [269].

#### 4.3.17. Treatment of Systemic Mycoses

Treatments for these severe systemic fungal infections include systemic azole agents (itraconazole, fluconazole, voriconazole, isavuconazole, and posaconazole), echinocandins (caspofungin, micafungin, anidulafungin, rezafungin), antimetabolites (flucytosine) or polyene therapy (liposomal amphotericin B). In most severe cases, a combination of antifungal agents is considered to seek synergy.

Some of these fungi are inherently non-susceptible to standard antifungal therapy, alternative antifungal agents, and surgical management may be required. Furthermore, it is crucial to reverse the underlying impairment of host defenses.

## 5. Prospects

Some predictions can be made concerning the evolution of the trends of fungal diseases in the decades ahead [273]. Climate change can act as a driver for changes in the patterns of fungal diseases [274]. For dry-weather fungal spores (e.g., *Alternaria* spp., *Cladosporium* spp., *Epicoccum* spp.), warmer temperatures at higher latitudes will contribute to the colonization of new geographical locations, increasing the distribution of otherwise endemic fungal pathogens. On the other hand, the increase in rainfall and relative humidity levels associated with floods and hurricanes will favor the increase in concentrations of wet weather spores (e.g., *Aspergillus* spp., *Didymella* spp., and *Penicillium* spp.).

The number of patients at risk is continuously increasing. Among these patients are those with HIV/AIDS and those subjected to chemotherapy or immunosuppression therapies, mainly in low-income countries with limited health resources [275]. The number of patients surviving malignancy, organ transplant, and autoimmune conditions is expected to continue to increase in the following decades [276]. Bearing these aspects in mind, financial support to low-income countries will improve the diagnosis and treatment of fungal diseases. In fact, in resource-limited settings, such as the African continent, the unavailability of timely and accurate diagnosis (e.g., point-of-care tests) and the high cost of laboratorial testing may delay the early diagnosis and initiation of appropriate antifungal therapy and, therefore, increase morbidity and mortality [277].

Despite all efforts, no vaccines or immune therapies for fungal diseases are available. The development of these products, their commercialization and distribution would reduce the number of fungal diseases worldwide.

Another matter of concern is the increasing number of antifungal resistances. So far, four main drug classes (i.e., polyenes, azoles, echinocandins, and triterpenoids) are available. Resistance to azoles, driven by the over-usage of fungicides for agricultural and veterinary purposes that later spread to human pathogens by several mechanisms (e.g., plasmids, transposons, lateral and horizontal gene flow), has already been reported in several fungal human pathogens, such as *A. fumigatus* and *Fusarium* spp. [278]. It is expected that, over time, the number of drug-resistant fungal strains will increase, highlighting the need to control the use of these drugs not only in human health but also in veterinary medicine and agriculture [279]. The increasing appearance of drug resistance also demonstrates the need for further research to develop new therapeutic approaches.

## 6. Conclusions

Although with different concentrations, fungal spores are an ever-present component of indoor and outdoor environments. These airborne particles can cause several human diseases ranging from mild allergic diseases (mainly associated with immunocompetent individuals) to severe disseminated infections (mainly associated with immunocompromised patients).

More than thirty fungal genera have been identified as producing allergens so far (Appendix A). These antigens are inciting agents of allergic diseases such as allergic bronchopulmonary aspergillosis, severe asthma with fungal sensitization, thunderstorm asthma, allergic fungal rhinosinusitis, and several occupational lung diseases.

Around sixty fungal genera are responsible for the onset of fungal infections (superficial, subcutaneous, and systemic infections). The main risk factors associated with the development of these diseases are underlying medical conditions mainly associated with a secondary immunodeficiency such as diabetes mellitus, the use of medication (broad spectra antibiotics, cancer treatments, corticosteroids, immunosuppressors), together with environmental exposure associated with either healthcare facilities, work, or leisure activities. Routes of transmission include inhalation, the entry of fungal spores through skin injuries, ingesting contaminated food products, nosocomial infections, and person-to-person transmission. The main clinical manifestations include respiratory (e.g., cough, dyspnea, chest pain) and central nervous system symptoms (e.g., mental confusion, behavioral changes, nausea). Currently, only four drug classes (i.e., polyenes, azoles, echinocandins, and triterpenoids) are used in therapeutics.

More profound knowledge of the diversity of fungal diseases will increase the awareness of these diseases, and the control of associated risk factors, improving its diagnosis and treatment.

## Figures and Tables

**Figure 1 jof-09-00381-f001:**
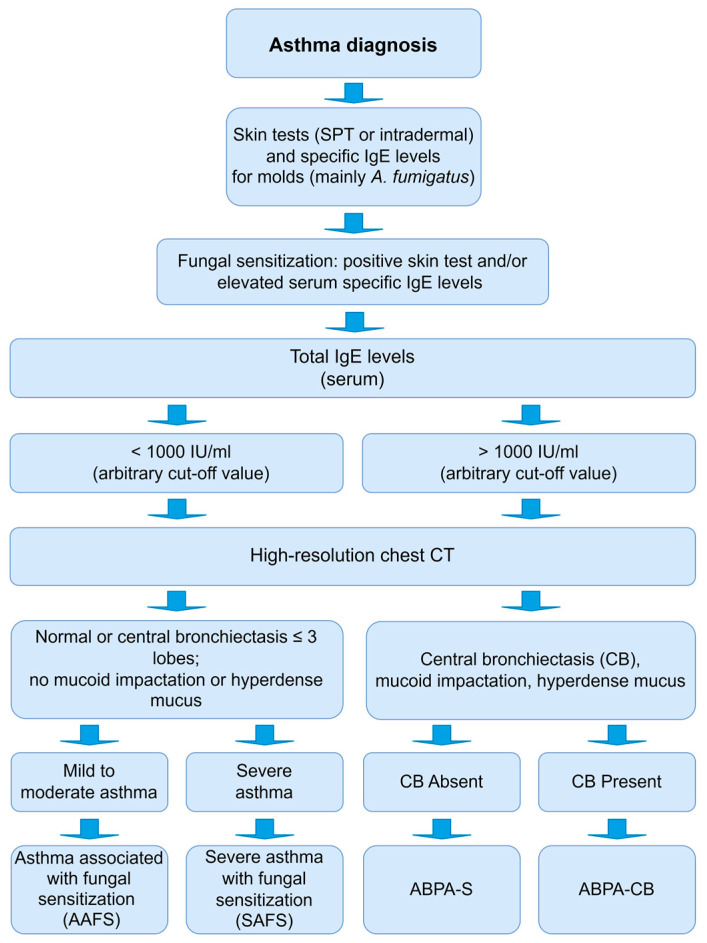
Diagnostic algorithm for SAFS versus ABPA (modified from [47]).

**Figure 2 jof-09-00381-f002:**
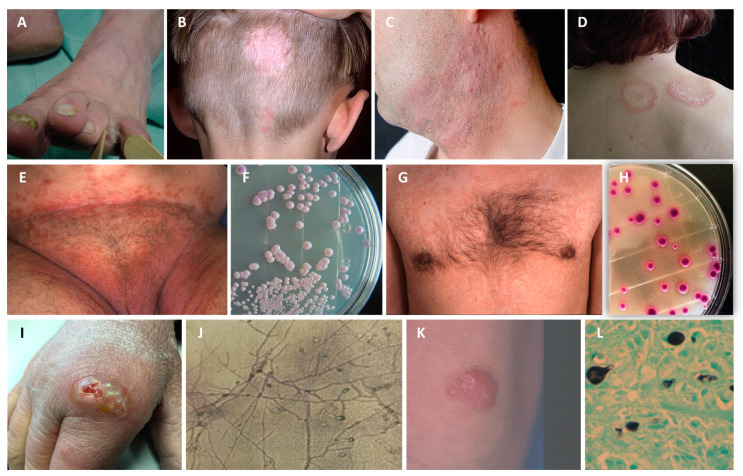
Clinical presentations and diagnosis of superficial and subcutaneous fungal infections (**A**): *tinea pedis* and onycomycosis; (**B**): *tinea capitis*; (**C**): *tinea barbae*; (**D**): *tinea corporis*; (**E**): candidiasis; (**F**). *Candida* isolate grown on CHROMagar; (**G**). *Pityriasis versicolor*; (**H**). *Malassezia* isolate on CHROMagar; (**I**): Subcutaneous fungal abscess in a renal transplant recipient; (**J**). *Scedosporium apiospermum* lactophenol cotton blue preparation showing conidia and septate hyphae; (**K**). Subcutaneous alternariosis in a renal transplant recipient with pulmonary *Alternaria* infection; (**L**): *Alternaria* spp. in tissue showing pigmented conidia (Gomori methenamine silver stain).

**Table 1 jof-09-00381-t001:** Overview of the most common allergic fungal diseases.

Disease	Etiological Agent	Risk Factors	Clinical Manifestations	Diagnosis	Treatment
Allergic Bronchopulmonary Aspergillosis	*A. fumigatus* *A. terreus* *A. niger* *A. flavus*	Asthma (2.5% asthmatic patients) Cystic fibrosis (1–15% of CF patients)	Worsening asthma Worsening cystic fibrosis Coughing paroxysms Mucoid impaction, with consolidation of the lung Recurrent ‘chest infections’	Total IgE > 1000 KIU/L SPT or specific IgE positive for Af Fleeting or fixed pulmonary opacities on chest radiograph Continuing respiratory symptoms Af IgG antibodies Eosinophilia (>500 cells/μL in steroid naïve patients) Filamentous fungal growth in sputum cultures or bronchial lavage fluid Sputum or bronchoscopy samples with positive PCR for Af	Oral and inhaled corticosteroids Adjuntive oral antifungal medication (itraconazole, voriconazole) Biologicals (omalizumab, mepolizumab, benralizumab, dupilumab)—off label
Severe Asthma with Fungal Sensitization	*A. fumigatus* *P. chrysogenum* *C. herbarum* *A. alternata* *C. albicans* *Trichophyton* spp.	Rarely any underlying disease	Worsening asthma symptoms Need for a high dose of inhaled steroids and/or frequent courses of oral steroids	Low FEV1 or peak flow (usually persistently) Total IgE < 1000 KIU/L SPT or specific IgE test positive for any fungus High-resolution chest CT: normal or minor alterations	Similar to severe asthma Partial or absent treatment response: consider biologicals (omalizumab, mepolizumab, benralizumab) and antifungal drugs (itraconazol)
Thunderstorm asthma	*Alternaria* spp. *Cladosporium* spp. *Diatrypaceae* *D. exitialis* *P. nigran* *Sporobolomyces* spp.	Sensitization and exposure to aeroallergens (mainly pollen grains and fungal spores) Asthma Allergic rhinitis Age Gender Ethnicity	The same as asthma	The same as asthma	The same as asthma
Allergic fungal rhinosinusitis	*A. fumigatus* *A. flavus* *B. spicifera* *C. lunata* *A. alternata* Other dematiaceous fungi	Atopy	Nasal obstruction Loss of smell Nasal discharge Nasal crust Pressure sensation over the face sinus Double vision or visual loss (rare) Facial asymmetry and proptosis Sino–bronchial allergic mycosis syndrome	SPT or elevated fungal-specific IgE Imagological findings (CT or MRI of the sinuses): opacification with centrally hyperdense content, nasal polyposis Histopathology showing eosinophilic mucin without invasion into the sinus tissue Fungal hyphae in the mucus	Topical and oral corticosteroids Saline douching Sinus surgery Biologicals (dupilumab, omalizumab, mepolizumab) Allergen immunotherapy (?) Antifungal therapy (?)

Af = *A. fumigatus*; SPT = Skin-prick test; CF = Cystic Fibrosis; CT = Computed Tomography; MRI = Magnetic Resonance Imaging; PCR = Polymerase Chain Reaction; SPT = Skin-Prick Test.

**Table 2 jof-09-00381-t002:** Overview of the most common occupational fungal diseases.

Disease	Etiological Agents	Risk Factors
Farmer’s lung	*Alternaria* spp. *A. fumigatus* *A. glaucus* *Botrytis* spp. *P. brevicompactum* *P. olivicolor*	Handling damp hay Opening bales for feeding livestock Threshing moldy grain
Mushroom worker’s lung	*A. bisporus* *H. tessellatus* *L. edodes* *P. ostreatus*	Exposure in spawning sheds
Suberosis	*P. glabrum* *C. sitophila*	Handling of damp cork
Maple bark stripper’s lung	*C. corticale*	Stripping bark from logs
Sequoiosis	*Graphium* spp. *Pullularia* spp. *A. pullulans*	Breathing damp sawmill dust
Wood pulp worker’s lung	*Alternaria* spp.	Pulping contaminated wood
Malt worker’s lung	*A. clavatus* *P. granulatum* *P. citrinum* *R. stolonifer*	Handling moldy grain
Wine grower’s lung	*B. cinerea*	Mold contamination
Baker’s lung disease	*A. fumigatus*	Handling contaminated flour
Cheese worker’s lung	*P. roqueforti* *P. casei* *P. notatum* *P. viridicutum* *A. fumigatus* *A. niger* *A. pullulans*	Cleaning mold off cheese
Salami Brusher’s disease	*P. glabrum* *P. camemberti* *P. nalgiovense* *A. fumigatus* *Cladosporium* spp.	Cleaning the white mold growing on salami surface using a manual wire brush
Tobacco worker’s lung	*A. fumigatus*	Exposure to tobacco dust and molds dispersed in the air in cigarette production facilities
Peak moss worker’s lung	*Monocillium* spp. *P. citreonigrum*	Handling contaminated peat moss
Paprika slicer’s lung	*M. stolonifer*	Handling moldy paprika pods during slicing
Summer-type HP	*T. cutaneum* *T. asahii* *T. mucoides*	Exposure to damp housing and furniture materials
Humidifiers, heating, and ventilation systems	*Aspergillus* spp. *Cladosporium* spp. *Penicillium* spp. *A. pullulans* *Cephalosporium* spp. *Mucor* spp. *Rhodoturula* spp.	Humidifier ran continuously without cleaning and with water added periodically

**Table 3 jof-09-00381-t003:** Primary Immunodeficiencies (PID) associated with chronic mucocutaneous candidiasis (adapted from [125,133]).

PID		Inheritance	CMCC Incidence	Other Fungal Infections	Non-Fungal Infections	Noninfectious Complications
Cellular and combined immunodeficiencies	SCID	X-linked or AR	30–35%	Variable (e.g., PCP)	Bacteria, viruses and mycobacteria	
	DOCK8 deficiency (AR-HIES)	AR	53–64%	Histoplasmosis Cryptococcosis PCP	Staphylococcal infections, HSV, VZV, HPV, molluscum contagiosum virus	Eczema, food allergy and asthma, malignancies (lymphoma)
	PGM3 deficiency (AR-HIES)	AR	Variable	Variable	HSV, CMV	Eczema, asthma, neurologic impairment, leukocytoclastic vasculitis
Defective Th17 immunity	STAT3 LOF mutation (AD-HIES)	AD	80%	Aspergillosis Cryptococcosis Histoplasmosis	*S. aureus*	Eczema, pneumatoceles, coronary aneurisms, hyperextensible joints
	STAT1 GOF mutation	AD	100%	Histoplasmosis Coccidioidomycosis	HSV	Autoimmunity Cerebral and aortic aneurisms
	CARD9 deficiency	AR	Variable	*Candida* meningoencephalitis Deep dermatophytosis	None	None
	APECED	AD or AR	90–100%	None	None	Hypoparathyroidism Adrenal insufficiency Vitiligo, alopecia, keratoconjunctivitis
	IL-17F mutation	AD	Variable	None	None	None
	IL-17RC mutation	AR	Variable	None	None	None
	IL-17RA and ACT1 mutations	AR	Variable	None	Staphylococcal infections (dermatitis and blepharitis)	None
	RORC mutation	AR	90–100%		Disseminated mycobacterial infection	None

Legend: AD: autosomal dominant; APECED: autoimmune polyendocrinopathy, candidiasis, ectodermal dystrophy; AR: autosomal recessive; CARD9: caspase recruitment domain-containing protein 9; CMCC: Chronic mucocutaneous candidiasis; CMV: Cytomegalovirus; GOF: gain-of-function; HIES: Hyper-IgE syndrome; HPV: human papillomavirus; HSV: Herpes simplex virus; LOF: loss-of-function; PCP: Pneumocystis pneumonia; PGM3: Phosphoglucomutase 3; SCID: Severe combined immunodeficiency; STAT3: Signal transducer and activator factor of transcription-3; VZV: Varicella zoster virus.

**Table 4 jof-09-00381-t004:** Primary immunodeficiencies (PID) and invasive fungal disease (adapted from [125,133]).

PID		Infections					
		Invasive Aspergillosis	Invasive Candidiasis	Dimorphic Fungi Infections	Cryptococcosis	*Pneumocystis jirovecii* Pneumonia	Deep Dermatophytosis
Phagocytic defects	CGD	High	Occasionally	-	-	-	-
	LAD-1	Occasionally	Occasionally	-	-	-	-
	Congenital neutropenia	Occasionally	Occasionally	-	-	-	-
Cellular and combined deficiencies	SCID, CIDs (e.g., AR-HIES, X-linked HIGM, IPEX)	Occasionally	Variable susceptibility (e.g., IPEX)	Variable susceptibility to Histoplasmosis, mostly in AR-HIES (DOCK8 deficiency) and HIGM	Variable, mostly in AR-HIES (DOCK8 deficiency) and HIGM	High in patients with X-linked HIGM syndrome; variable in patients with SCID, AR-HIES, NEMO and MHC class II deficiency	-
Defective Th17 immunity	AD-HIES (STAT3)	High in patients with lung cavities	Occasionally	Histoplasmosis Coccidioidomycosis	Occasionally	-	-
	STAT1 GOF mutation	.	-	Coccidioidomycosis Histoplasmosis	-	-	-
	CARD9 deficiency	.	Invasive CNS candidiasis	-	-	-	High

Legend: AR: autosomal recessive; CARD9: caspase recruitment domain-containing protein 9; CGD: Chronic granulomatous disease CID: Combined immunodeficiency CNS: Central nervous system; HIES: Hyper-IgE syndrome HIGM: Hyper-IgM syndrome; IPEX: Immune dysregulation, polyendocrinopathy, and enteropathy, X linked; LAD: Leukocyte adhesion deficiencies; NEMO: Nuclear factor-kB essential modulator; SCID: Severe combined immunodeficiency; STAT1: Signal transducer and activator factor of transcription-1.

## Data Availability

Data sharing not applicable.

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
