# Peer review of "Clinical Manifestations of Human Exposure to Fungi"

_jof, 2023, doi:10.3390/jof9030381_

Round 1

Reviewer 1 Report

The article by Manuela Oliveira et al. reviews the implications of fungi in human health from allergic diseases to infections. Reviewing topics such as clinical manifestations, etiological agents, risk factors to improve the knowledge of this growing health problem.

From my point of view the manuscript is well written, easy to follow and the subject is interesting because the authors make a very complete and updated review of the actual situation of diseases or pathologies caused by fungi. Perhaps because of it, being so complete, I think the tables presented are too dense.

However, I think that some issue must be addressed before the article can be fully considered for publications.

Some comments/suggestions I can make are:

-           - Table 1. The authors could to try somehow to modify it so that it is not so arduous. Define please the format well, the rows are not delimited

-          - Tables 2 and 3, could be unified as primary inmunodeficiences associated with fungi disease

-          - Table 4, it is too long. It could be included as an annexe

-          - P. 9. The text is horizontally oriented

-          -  Line 77. A. fumigatus is not in cursive

-          - Unify the abbreviations throughout the text, put them or not in the titles of the different sections. In some it appears and in others it does not. Once the abbreviation is indicated, use it every time it is mentioned in the text. Line 183, line 209…

-         - Section 2.5 Occupational lung disease would recommend a schematic Table

-     - It would be nice in the text, when talking about diagnosis, to mention the available commercial tests that exist.

-         - In paragraph, lines: 1175- 1181,… the use in agriculture of products such as pesticides, etc. is mentioned, but I would recommend expanding it due to the current imminent problem of treatment due to the great increase in resistant A. fumigatus strains that are appearing due to this problem, just as it will be in others fungiI suggest including an article about this.

Author Response

Comments and Suggestions for Authors (Reviewer 1)

The article by Manuela Oliveira et al. reviews the implications of fungi in human health from allergic diseases to infections. Reviewing topics such as clinical manifestations, etiological agents, and risk factors to improve the knowledge of this growing health problem.

From my point of view, the manuscript is well written and easy to follow, and the subject is interesting because the authors make a very complete and updated review of the actual situation of diseases or pathologies caused by fungi. Perhaps because it, is so complete, I think the tables presented are too dense.

However, I think that some issues must be addressed before the article can be fully considered for publication.

Some comments/suggestions I can make are:

  • Table 1. The authors could try somehow to modify it so that it is not so arduous. Define please the format well, the rows are not delimited.

Thanks for these suggestions. Accordingly, we decided to split Table 1, removing data referring to occupational diseases, and creating a new Table (now, Table 2) with data on occupational fungal lung diseases, also for comment 1.7.

Please note that some small modifications were introduced in the text of Table 1, improving its readability.

  • Tables 2 and 3, could be unified as primary immunodeficiencies associated with fungi disease

Thank you for the note. Regarding the unification of former Tables 2 and 3 (now, Tables 3 and 4), please note that despite being in the same section (3. Association of immunodeficiencies and fungal infections), each Table addresses different clinical presentations in distinct primary immunodeficiencies: Table 3 focus on mucocutaneous presentations of the immunodeficiencies associated with chronic mucocutaneous candidiasis and Table 4 focus on systemic invasive fungal diseases. For this reason, we would like to consider keeping both Tables, which are quite informative on their specific topic.

  • Table 4, it is too long. It could be included as an annex.

Thank you for the suggestion. The Table is now removed and renamed as Table S2

  • 9. The text is horizontally oriented.

Text orientation changed.

  • Line 77. fumigatus is not in cursive.
  1. fumigatus was written in italics.

  • Unify the abbreviations throughout the text, put them or not in the titles of the different sections. In some, it appears and in others, it does not. Once the abbreviation is indicated, use it every time it is mentioned in the text. Line 183, line 209…

Thanks for the comment. The use of abbreviations in the titles was unified. After being defined, the abbreviation was used throughout the text, except when it appears in the beginning of a sentence.

  • Section 2.5 Occupational lung disease would recommend a schematic Table.

Accordingly, we created a new Table (now Table 2) with data on occupational fungal lung diseases that were removed from Table 1 (also per the comment 1.1).

Please note that the schematic table on occupational lung diseases includes 3 columns: disease, etiological agent, and risk factors.

  • It would be nice in the text, when talking about diagnosis, to mention the available commercial tests that exist.

We understand your point of view, but please note that availability of commercial tests varies widely from country to country as commercial brands. 

  • In paragraph, lines: 1175 – 1181, the use in agriculture of products such as pesticides, etc. is mentioned, but I would recommend expanding it due to the current imminent problem of treatment due to the great increase in resistant fumigatus strains that are appearing due to this problem, just as it will be in others fungi. I suggest including an article about this.

Text (lines 1232 to 1235) changed accordingly to “Resistance to azoles, driven by the over-usage of fungicides for agricultural and veterinary purposes that latter spread to human pathogens by several mechanisms (e.g., plasmids, transposons, lateral and horizontal gene flow), has already been reported in several fungal human pathogens, such as A. fumigatus and Fusarium spp. [279].” 

Reviewer 2 Report

The authors have presented an extensive list of fungal pathogens, while including a reasonable numbers of features of each group. There are new diseases described that I have now read about, even after 45 years of teaching/research in Medical Mycology.

Below I list suggestions for changes that may be useful to readers. There are very few grammatical changes, so I believe the manuscript has been well-edited by the authors.

1.    I did not understand the page numbering sequence which changes throughout the entire manuscript.

2.    I was under the impression that blastomycosis is found more north in the US for example, Wisconsin, etc.

3.    I think an interesting well-described observation is the delay in exposure and disease symptoms typically seen with cocci and blasto. Perhaps statements of epidemics should also be included in a collective manner.

4.    Prospects section. Climate change is mentioned but unreferenced even though there is published data. Go to pubmed and list key words of climate change/fungi. In fact, there are no references in the Prospects section.

5.    Prospects. Global sites that reflect differences in incident include standard of living. Crypto in the US versus some south africian countries. There is literature on this subject.

6.    While an extensive explanation of antifungals and resistance is not needed, there is relatively new literature that links A. fumigatus and Fusarium species with environmentally, acquired azole resistance. Fungal plant diseases are treated with triazoles. Spore disssemination of resistant strains are acquired by humans through inhalation and cause hospital acquired diseases.  This has led to the the definition of these fungi as Trans-Kingdom Pathogens (Plants and Humans). 

Author Response

Comments and Suggestions for Authors (Reviewer 2)

The authors have presented an extensive list of fungal pathogens while including a reasonable number of features of each group. There are new diseases described that I have now read about, even after 45 years of teaching/research in Medical Mycology.

Below I list suggestions for changes that may be useful to readers. There are very few grammatical changes, so I believe the manuscript has been well-edited by the authors.

  • I did not understand the page numbering sequence which changes throughout the entire manuscript.

The authors acknowledge this observation. The page number sequence was corrected.

  • I was under the impression that blastomycosis is found more north in the US for example, Wisconsin, etc.

The text (lines 856-860) was changed accordingly, and two new references were added: “In North America, the area of endemicity includes the southeastern and south-central states, especially those bordering the Ohio and Mississippi River basins; the Midwest states and Canadian provinces bordering the Great Lakes; and an area in New York and Canada along St. Lawrence Rivera. Blastomycosis is also endemic in Africa [222, 223].”

Lee, P.P.; Lau, Y.L. (2017). Cellular and molecular defects underlying invasive fungal infections-revelations from endemic mycoses. Frontiers in Immunology 8, 735.

Murray, P. R.; Rosenthal, K. S.; Pfaller, M. A. (2020). Systemic Mycoses Caused by Dimorphic Fungi-Blastomycosis. In: Medical Microbiology, Elsevier, 634-36

  • I think an interesting well-described observation is the delay in exposure and disease symptoms typically seen with cocci and blasto. Perhaps statements about epidemics should also be included in a collective manner.

The text (lines 880-895) was changed accordingly: “As for most endemic mycoses, symptoms severity and disease course depend on exposure extent (inoculum size) and the immune status of the exposed individual. Based mainly on blastomycosis outbreak studies, the symptomatic disease occurs in less than half of infected individuals. Clinical illness caused by B. dermatitidis may present as pulmonary or extrapulmonary disseminated disease.

A classic form of blastomycosis is chronic cutaneous involvement that is almost always the result of hematogenous dissemination from the lung, in most instances without evident pulmonary lesions or systemic symptoms. Pulmonary blastomycosis may be asymptomatic or present as a mild flulike illness. More severe infection resembles bacterial pneumonia with acute onset, high fever, lobar infiltrates, and cough. Progression to fulminant adult respiratory distress syndrome with high fever, diffuse infiltrates, and respiratory failure may occur. A more subacute or chronic respiratory form of blastomycosis may resemble tuberculosis or lung cancer, with the radiographic presentation of pulmonary mass lesions or fibronodular infiltrates.

Clinical suspicion is essential for diagnosis, and delays can be avoided based on laboratory testing.”

  • Prospects section. Climate change is mentioned but unreferenced even though there is published data. Go to PubMed and list keywords of climate change/fungi. In fact, there are no references in the Prospects section.

References (274 to 279) were added to the Prospects section.

  • Global sites that reflect differences in incident include standard of living. Crypto in the US versus some south African countries. There is literature on this subject.

Text (lines 1222-1226) were changed accordingly to “In fact, in resource-limited settings, such as the African continent, the unavailability of timely and accurate diagnosis (e.g., point-of-care tests) and the high cost associated with the high cost of laboratory testing may delay the early diagnosis and initiation of appropriate antifungal therapy and, therefore, increase morbidity and mortality [278].”

  • While an extensive explanation of antifungals and resistance is not needed, there is relatively new literature that links  fumigatusand Fusarium species with environmentally, acquired azole resistance. Fungal plant diseases are treated with triazoles. Spore dissemination of resistant strains is acquired by humans through inhalation and causes hospital-acquired diseases.  This has led to the definition of these fungi as Trans-Kingdom Pathogens (Plants and Humans).

Text (lines 1232 to 1235) changed accordingly to “Resistance to azoles, driven by the over-usage of fungicides for agricultural and veterinary purposes that later spread to human pathogens by several mechanisms (e.g., plasmids, transposons, lateral and horizontal gene flow), has already been reported in several fungal human pathogens, such as A. fumigatus and Fusarium spp. [278].”

Reviewer 3 Report

The manuscript “Clinical manifestations of human exposure to fungi” was reviewed. Although there are several published reviews on this topic, this work has gathered a lot of information about different fungal infections. However, I can recommend that to remove the repeating data, such as ABPA, superficial and cutaneous fungal infections. Prepared tables are very useful, so I recommend keeping them in the manuscript. An interesting part of this manuscript is the prepared data on immunodeficiencies and invasive fungal diseases, which I would recommend the authors to focus on instead of repeating the mentioned information.

Fungal names are changed to italic format throughout the manuscript (especially lines 701-707).

Author Response

Comments and Suggestions for Authors (Reviewer 3)

The manuscript “Clinical manifestations of human exposure to fungi” was reviewed. Although there are several published reviews on this topic, this work has gathered a lot of information about different fungal infections.

3.1. However, I can recommend that to remove the repeating data, such as ABPA, and superficial and cutaneous fungal infections.

3.2. Prepared tables are very useful, so I recommend keeping them in the manuscript. An interesting part of this manuscript is the prepared data on immunodeficiencies and invasive fungal diseases, which I would recommend the authors focus on instead of repeating the mentioned information.

Thanks for the positive comments (3.1 and 3.2), which we will answer together. This manuscript is a narrative review of an extensive field encompassing diverse clinical manifestations of human exposure to fungi. For this reason, we addressed multiple human diseases, from hypersensitivity reactions, to immunodeficiencies, and fungal infections. Within this context, ABPA (Table 1 and Section 2.1) and superficial and subcutaneous fungal infections (Table S2, Sections 4.1 and 4.2) are common presentations, of fundamental clinical relevance, and are not repeated elsewhere in the manuscript. For this reason, we would like to consider keeping such sections of the text.

3.3. Fungal names are changed to italic format throughout the manuscript (especially lines 701-707).

Noted and corrected across the manuscript.